# High-Loaded Copper-Containing Sol–Gel Catalysts for Furfural Hydroconversion

**DOI:** 10.3390/ijms24087547

**Published:** 2023-04-19

**Authors:** Svetlana Selishcheva, Anastasiya Sumina, Evgeny Gerasimov, Dmitry Selishchev, Vadim Yakovlev

**Affiliations:** Boreskov Institute of Catalysis, Lavrentiev Ave. 5, Novosibirsk 630090, Russia; sumina@catalysis.ru (A.S.); gerasimov@catalysis.ru (E.G.); selishev@catalysis.ru (D.S.); yakovlev@catalysis.ru (V.Y.)

**Keywords:** hydroconversion, furfural, furfuryl alcohol, 2-methylfuran, Cu-containing catalyst

## Abstract

In this study, the high-loaded copper-containing catalysts modified with Fe and Al were successfully applied for the hydroconversion of furfural to furfuryl alcohol (FA) or 2-methylfuran (2-MF) in a batch reactor. The synthesized catalysts were studied using a set of characterization techniques to find the correlation between their activity and physicochemical properties. Fine Cu-containing particles distributed in an amorphous SiO_2_ matrix, which has a high surface area, provide the conversion of furfural to FA or 2-MF under exposure to high pressure of hydrogen. The modification of the mono-copper catalyst with Fe and Al increases its activity and selectivity in the target process. The reaction temperature strongly affects the selectivity of the formed products. At a H_2_ pressure of 5.0 MPa, the highest selectivity toward FA (98%) and 2-MF (76%) was achieved in the case of 35Cu13Fe1Al-SiO_2_ at the temperature of 100 °C and 250 °C, respectively.

## 1. Introduction

Due to a few energy and environmental issues, it is important to find alternative sources of raw materials to produce biofuels, valuable chemicals, and various fuel additives. Such a source can be lignocellulose, which is characterized by a low content of harmful elements (heavy metals, as well as sulfur and nitrogen, resulting in emissions of NO_x_ and SO_x_) and has large volumes of annual production. Hemicellulose is one of the main components of lignocellulose, which can be converted into furfural by acid hydrolysis. Furfural is included in the list of the 30 most biomass-derived valuable chemicals [1,2] and has a wide range of applications. Up to 80 value-added chemicals can be obtained as a result of furfural conversion in various pathways [3,4,5]. Thus, furfural can be converted into various valuable C_4_–C_5_ chemicals such as furfuryl alcohol (FA), valerolactone, pentanediols, cyclopentanone, dicarboxylic acids, butanediol, and butyrolactone [6]. Additionally, furfural can be selectively hydrogenated into potential high-octane additives, such as 2-methylfuran (2-MF, sylvan), or can undergo a combination of aldol condensation, esterification, and hydrodeoxygenation to form liquid alkanes [7,8,9]. The 2-MF can be used in existing internal-combustion engines as a fuel additive [10,11,12,13] due to its high octane number (131) and good volatility (boiling point of 65 °C). Corma et al. demonstrated the possibility of obtaining branched alkanes from 2-MF for diesel fraction [14,15]. The 2-MF is used as a solvent in various processes, as a reagent in the production of pesticides, and as chloroquine phosphate for antimalarial drugs [16]. However, ca. 65–85% of furfural is used to produce FA [17,18,19]. FA is widely used as a solvent in various processes, as a reagent in the production of foundry resins and furan fiber-reinforced plastics, drugs (e.g., ranitidine), flavors, 2-MF, 2-methyltetrahydrofuran, and tetrahydrofurfuryl alcohol [3,20,21,22]. Additionally, FA is a very important substance for the production of vitamin C, lysine, and lubricants [23,24]. Therefore, special attention is given to the selectivity of catalysts in the hydrogenation of furfural because FA is an intermediate compound in the chain of furfural transformation. Additionally, furfural is used to produce furan resins and selectively purify lubricating oils, to synthesize many commercial products (e.g., tetrahydrofurfuryl alcohol (THFA), furan, tetrahydrofuran (THF)).

Conventional catalysts in the production of FA from furfural are the CuCr catalysts, which provide a high yield of FA (up to 98%) on an industrial scale [25,26,27]. However, the main disadvantage of the CuCr system is the presence of chromium in its composition, which can contaminate the target products with chromium species. In addition, the coating of Cu active sites with chromium-containing particles formed from copper chromite under process conditions significantly reduces the selectivity of FA formation [28,29]. One of the ways to improve the stability of copper chromite is the atomic layer deposition of alumina, which prevents the formation of coke, sintering, and blocking of copper particles [28]. However, this technology is quite expensive and, for this reason, cannot be applied on an industrial scale. The systems based on noble metals (e.g., Pt, Pd) are also active in the hydroconversion of furfural, but they commonly have high costs and provide insufficient selectivity due to the formation of products from the hydrogenation furan ring [30,31,32]. Therefore, the development of a highly active, selective, and eco-friendly catalyst for the hydroconversion of furfural is an important task. It is important to note that the efficient catalyst should provide selective hydrogenation of the aldehyde functional group in furfural, while the hydrogenation of the aromatic ring should be suppressed.

Cu-containing systems are the most promising catalysts for selective hydrogenation of furfural to FA or 2-MF. However, a monometallic copper catalyst does not have sufficient activity in the target process because of the accumulation of carbon on the surface of the catalyst and the agglomeration of the active component. High values of hydrogen pressure are commonly used to solve this problem [33]. Another way to increase the activity of copper catalysts is the addition of modifiers (e.g., Fe, Mo, Al, Ni, Co, Zn) [34,35,36,37,38,39,40]. However, it is not possible in many cases to achieve the required results due to strong adsorption of reaction products on the catalyst surface (thus providing a decrease in the selectivity toward FA), changes in the oxidation state of the active component, and polymerization of furfural with itself and FA molecules [33,41].

The catalysts based on iron oxide have activity in the oxidative processes, for example, esterification of alcohols and aldehydes [42], decarbonylation of aliphatic aldehydes [43], ammoxidation of aromatic aldehydes [44], and heavy oil cracking in the presence of steam [45]. Ma et al. [46] showed the possibility of using magnetic Fe_3_O_4_ nanoparticles in the catalytic hydrogenation of furfural to FA. It was concluded that the smaller Fe_3_O_4_ particles had the larger BET surface area and pore diameter, thus providing better catalytic performance (FA yield of 90% at furfural conversion of 97%, 160 °C, 5 h, 0.1 g of catalyst, 0.67 mmol of furfural, 20 mL of alcohol). Alumina is known to have Lewis acid sites and to promote the activation of C=O bonds in aldehydes or ketones [47]. On the other hand, the Lewis acid sites can contribute to the formation of carbon deposits on the surface, thus providing additional stability of the catalyst to the formation of secondary products under reaction conditions. Thus, the use of copper-containing catalysts modified with iron and aluminum oxides can increase the catalytic performance and provide high selectivity to the formation of FA under reaction conditions.

In this work, homophase and heterophase preparation techniques were used for the synthesis of Cu-containing catalysts to provide selective hydrogenation of furfural [48,49]. Both techniques are based on the sol–gel method using partially hydrolyzed tetraethoxysilane (ethyl silicate, ES), which has a wide range of applications as a binder and hydrophobic agent. Dispersion of the metal precursor in an alcohol solution containing ES and subsequent drying lead to the condensation of ES with the formation of a polysiloxane film directly on the surface of the precursor. The next steps of calcination and reduction result in the formation of a SiO_2_ matrix with dispersed metal particles enclosed in its pores. The formed matrix prevents the sintering of metal particles during the reduction and heat treatments.

## 2. Results and Discussion

### 2.1. Furfural Hydroconversion in the Presence of Monometallic Copper Catalysts

The monometallic copper catalysts with Cu content of 5–50 wt.% were investigated in the hydroconversion of furfural in a batch reactor. For all samples, only furfuryl alcohol (FA) was detected as a product of this reaction. Thus, the monometallic copper catalysts showed 100% selectivity to the formation of FA. The 5Cu-SiO_2_ catalyst was the least active, with a furfural conversion of 19% (Figure 1). The low activity of the 5Cu-SiO_2_ catalyst is probably due to insufficient content of the active component. The samples with a copper content of 10–30 wt.% showed similar activity in the process, while the yield of the target product at the end of the reaction was 82–85%. The most active catalysts in the hydrogenation of furfural were the samples with copper contents of 15 and 40 wt.%. They provided FA yields of 92 and 93%, respectively.

Table 1 summarizes the FA yield and the amount of CO chemisorbed by the reduced monometallic xCu–SiO_2_ catalysts at 200 °C. The amount of CO chemisorbed by catalyst samples reflects the number of accessible Cu sites. Although the heat of CO adsorption is extremely low in the case of copper compared, for example, with nickel [50], the measurement of this parameter allows us to draw some conclusions related to the activity of copper-containing systems [51,52,53].

CO chemisorption data have a good correlation with FA yields for all studied catalysts. A decrease in the activity, as well as a decrease in the amount of chemisorbed CO for 50Cu-SiO_2_ compared to 40Cu-SiO_2,_ can be explained by an increase in the particle size of the active component on the surface, which is confirmed by XRD and BET analyses (see Table 2).

The most active catalysts in this series are 15Cu-SiO_2_ (HomSG series) and 40Cu-SiO_2_ (HetSG series); however, it is clear from the kinetic plots presented in Figure 1 that after 50 min of reaction, there is no increase in the conversion of furfural until the end of the experiment. One of the approaches to increase the activity of copper catalysts is the introduction of modifying additives (e.g., Fe, Mo, Al), which make it possible to increase the catalytic performance of copper systems by preventing carbon accumulation on the catalyst surface and agglomeration of the active component due to the formation of more dispersed particles. Thus, two samples, namely, 15Cu-SiO_2_ and 40Cu-SiO_2_, were selected from a series of mono-copper catalysts for further modification and study in the process of furfural hydro conversion.

### 2.2. Furfural Hydroconversion in the Presence of Copper-Containing Catalysts Modified with Iron and Aluminum

Figure 2 shows the dependence of FA yield on the reaction time in the presence of copper–iron catalysts with different Cu/Fe molar ratios prepared by the homogeneous sol–gel method (15Cu-SiO_2_ sample was used as the basis). The modification of the 15Cu-SiO_2_ sample with iron does not lead to an increase in the activity of the initial Cu-catalyst but also reduces the FA yield to 70–77%. Apparently, small particles of the active component (CSR of 40 Å, XRD data, Table 2) are shielded by modifier particles (even in the case of a high Cu/Fe molar ratio), which leads to a decrease in the activity of the catalysts. Additionally, according to the XRD analysis (Table 2), the reflections corresponded to CuO disappear in the 15Cu5Fe-SiO_2_ catalyst compared to 15Cu-SiO_2_, which indicates a decrease in the size of these particles due to the presence of X-ray amorphous iron oxide.

The copper–iron catalysts prepared by the heterophase sol–gel method (40Cu-SiO_2_ was used as the basis) were also studied in the hydrogenation of furfural to FA. Figure 3 shows the dependence of FA yield on the reaction time in the presence of copper–iron catalysts with different Cu/Fe molar ratios. The modification of 40Cu-SiO_2_ with iron, in some cases, leads to an increase in activity. This is especially noticeable for the 35Cu13Fe-SiO_2_, 40Cu7Fe-SiO_2_, and 40Cu20Fe-SiO_2_ samples. These catalysts make it possible to obtain a higher FA yield (97–99%) under equal process conditions. The 35Cu13Fe-SiO_2_ (Cu/Fe = 2.4) turned out to be the catalyst, which completely converted furfural into FA, and the yield of FA was 99%.

Thus, the incorporation of iron into a high-percentage copper catalyst prepared by the heterogeneous sol–gel method makes it possible to increase the FA yield due to the formation of finer particles of the active component, which are more active in the target process. This assumption is confirmed by the data of XRD analysis (Table 2), which show only traces of crystalline copper oxide, while the major part of the oxide is X-ray amorphous. However, this fact causes an increase in the formation of carbon deposits compared to the pristine catalyst, as will be discussed in Section 2.3.

As stated in Introduction, the Lewis acid sites of alumina can promote the formation of carbon deposits on their own surface, thus providing additional stability of the catalyst to the formation of secondary products under reaction conditions. In this case, an attempt was made to modify the copper–iron catalyst with aluminum with different Fe/Al molar ratios. The dependence of FA yield on the reaction time in the presence of CuFeAl-SiO_2_ catalysts with different Fe/Al ratios is shown in Figure 4.

The modification with aluminum (i.e., 35Cu13Fe1Al-SiO_2_ sample) made it possible to completely convert furfural into FA in a shorter time compared to the copper–iron catalyst. Additionally, the heterophase sol–gel method, used for the synthesis of this catalyst, and modification with iron and aluminum contribute to easier shaping compared pristine sample. This fact is an important advantage for the processes in fixed-bed reactors.

Next, the reaction temperature of furfural hydrogenation was varied in the presence of the 35Cu13Fe1Al-SiO_2_ catalyst at a hydrogen pressure of 5.0 MPa. The distribution of formed products is shown in Figure 5.

For the 35Cu13Fe1Al-SiO_2_ catalyst, a temperature of 100–140 °C is optimal to produce FA from furfural because FA selectivity achieves 97–98% at 99% conversion of furfural. At a temperature of 160 °C, FA selectivity is 96% due to the formation of 2-MF. The selectivity toward 2-MF at this temperature is 4%. The hydrogenation of the FA hydroxyl group leads to a more significant formation of 2-MF up to 5–25% at 180–200 °C (at 5 h). The distribution of products changes significantly at 250 °C, when the main product becomes 2-MF with a selectivity up to 76% at 100% conversion of furfural, while 2-MTHF (12% of selectivity) is also formed (Figure 6).

Thus, the 35Cu13Fe1Al-SiO_2_ catalyst is highly active and selective in the formation of FA from furfural in a batch reactor at 100–140 °C, a hydrogen pressure of 5.0 MPa, catalyst loading of 0.3–1.0 g, and a reaction time of 2.5–5.0 h. At 200–250 °C, the main product is 2-MF with a selectivity of 76%.

### 2.3. Catalyst Characterization

The phase composition and textural characteristics were determined for all catalysts in oxide form (see Table 2). All samples are characterized by hysteresis of the H3 type with slit-like pores between particles; the samples are predominantly meso-macroporous. The introduction of iron increases the surface area of the catalysts (HetSG series) and prevents the crystallization of copper oxide, which is confirmed by XRD data (it is not possible to determine the CSR of copper oxide due to its X-ray amorphism) (Table 2). Further addition of aluminum slightly reduces the surface area of the catalysts, but the XPS method determines the Cu state attributed to copper(II) oxide, which allows us to propose that the presence of aluminum promotes the crystallization of CuO.

X-ray diffraction patterns obtained by XRD for the most active 35Cu13Fe-SiO_2_, 35Cu13Fe1Al-SiO_2_, and 40Cu-SiO_2_ catalysts are shown in Appendix A. After the reaction, broad peaks appear at 2θ = 36.7°, 42.4°, and 61.8°, attributed to Cu_2_O [PDF 5–667], and at 2θ = 43.4° and 50.4°, attributed to the metallic Cu [PDF 4–836]. No Fe-containing phases were found, as in the case of the fresh catalysts (Table 2).

Thus, the phase composition of the catalysts after the reaction is represented by the metallic copper and prereduced Cu_2_O (Table 3). When iron is introduced into the 40Cu-SiO_2_ catalyst, the size of the metallic copper CSR does not change. However, it increases when aluminum is additionally introduced. The introduction of Al also increases the degree of copper oxide reduction: 50% and 75% of Cu are observed in the 35Cu13Fe-SiO_2_ and 35Cu13Fe1Al-SiO_2_ samples after the reaction, respectively.

The study of some catalysts after the reaction was carried out using X-ray photoelectron spectroscopy (XPS). We studied the most active samples: 35Cu13Fe1Al-SiO_2_ (fresh) and after the reaction, 15Cu-SiO_2_, 40Cu-SiO_2_, 50Cu-SiO_2_, 15Cu5Fe-SiO_2_, 35Cu13Fe-SiO_2_, and 35Cu13Fe1Al-SiO_2_. Relative concentrations of elements (i.e., atomic ratios) in the near-surface layer are shown in Table 4. The binding energies of Si2p, C1s, Cu2p_3/2_, Fe2p_3/2_, and O_1s_ are given in Appendix A. Due to the low concentration of aluminum and the overlap of the Al2p spectrum with the copper Cu3p spectrum, the surface concentration of aluminum cannot be determined by XPS.

With an increase in Cu content in the monometallic catalysts, the [Cu]/[Si] surface ratio increases; with the introduction of iron, this ratio slightly decreases due to the release of iron to the surface. After the reaction, the [Cu]/[Si] ratio decreases for the 35Cu13Fe1Al-SiO_2_ catalyst due to an increase in [Fe]/[Si] and carbon formation. At the same time, the metallic copper is not identified by XPS in the 35Cu13Fe1Al-SiO_2_ catalyst after the reaction, apparently due to its coating with oxides (HRTEM data are given below to confirm this fact). In the fresh 35Cu13Fe1Al-SiO_2_ catalyst, metallic copper is not detected because this sample is in the oxide form.

In the series of 15Cu-SiO_2_—40Cu-SiO_2_—50Cu-SiO_2_, the content of surface carbon increases (Table 4), which is in good agreement with the results of the CHNS analysis of these samples after the reaction (Table 5). The addition of aluminum to the 35Cu13Fe-SiO_2_ catalyst slightly reduces the carbon content on the surface, which also agrees with the results of the CHNS analysis (Table 5).

The Si2p spectra of the studied catalysts show a broad symmetrical peak corresponding to silicon in the Si^4+^ state (Appendix A). This peak was used as an internal standard (E_b_ = 103.3 eV) to consider the effect of sample charging. For silicon in the SiO_2_ structure, the Si2p binding energies are in the range of 103.3–103.8 eV [54,55,56].

Figure 7 shows the Cu2p spectra of the catalysts. Due to the spin-orbit splitting, the Cu*2p* spectra exhibit two groups of peaks related to the Cu2p_3/2_ and Cu2p_1/2_ levels, the integrated intensities of which are related as 2:1. The spectrum of the studied catalysts exhibits peaks with Cu2p_3/2_ binding energies in the region of 932.5 and 935.1 eV, as well as peaks in the region of 941.1–943.4 eV—X-ray satellites corresponding to a peak in the region of 935.1 eV.

The shape of the spectra allows us to state that part of the copper is in the Cu^2+^ state in the near-surface layer of the catalysts. Indeed, a characteristic difference between the Cu2p spectra of Cu(II) compounds is the high binding energies of Cu2p_3/2_ in the range of 933.6–935.3 eV and the presence of intense core-level satellites in the region of high binding energies [57]. For example, the integrated intensity of the core-level satellite in the CuO spectrum reaches 55% of the intensity of the main Cu2p_3/2_ line [58]. For copper in the metallic state and Cu(I) compounds, the Cu2p_3/2_ binding energy is in the range of 932.4–932.9 eV, while in the spectrum of metallic copper, there are no core-level satellite peaks. The intensity of the core-level peaks in the spectrum of Cu^1+^ does not exceed 15% of the intensity of the main Cu2p_3/2_ line. The peak at 932.5 eV can be attributed to both metallic copper and copper in the Cu^1+^ state. In the literature, for metallic copper, the Cu2p_3/2_ binding energies are given in the range of 932.5–932.6 eV; for Cu_2_O, the binding energy is in the same range as for the metallic state of copper. Since the Cu2p_3/2_ binding energies for Cu^0^ and Cu^1+^ are similar, the identification of copper states is a difficult task. The so-called Auger parameter α is usually used to determine the state of copper in such cases. This parameter is equal to the sum of the Cu2p_3/2_ binding energy and the position of the maximum of the CuLMM Auger spectrum on the electron kinetic energy scale [59]. In accordance with the literature data, the Auger parameters for bulk samples of metallic copper, Cu_2_O, and CuO are 1851.0–1851.4, 1848.7–1849.3, and 1851.4–1851.7 eV, respectively. For the 15Cu-SiO_2_ and 15Cu5Fe-SiO_2_ catalysts, it is not possible to measure the value of the Auger parameter due to the low surface concentration of copper in the analysis zone. The Auger parameter for other catalysts for the peak in the region of 932.5 eV is 1849–1849.3 eV, which corresponds to copper in the Cu^1+^ state (Table 4). For a fresh 35Cu13Fe1Al-SiO_2_ catalyst, the Auger parameter indicates that the copper is in the Cu^2+^ state (Appendix A). It is likely that in 15Cu-SiO_2_ and 15Cu5Fe-SiO_2_ copper is also in the Cu^1+^ state.

The Fe2p spectra of the studied catalysts are shown in Figure 8. As seen, the Fe2p spectra are a Fe2p_3/2_–Fe2p_1/2_ doublet, the integral intensities of which are in the ratio of 2:1. To determine the state of iron, both the position of the main Fe2p_3/2_ line and the shape of the Fe2p spectrum (intensity and relative position of the lines of core-level satellites due to the manifestation of many-electron processes) are also used. The position and intensity of the line of core-level satellites depend on the chemical state of iron. In the case of the studied catalysts, the spectra of Fe2p_3/2_ represent a peak with binding energy in the region of 711.6–712.3 eV, while core-level satellites are observed. In accordance with the literature data, iron in FeO, Fe_3_O_4_, and Fe_2_O_3_ is characterized by Fe2p_3/2_ binding energies in the ranges of 709.5–710.2, 710.1–710.6, and 710.7–711.2 eV, respectively [60,61], while the core-level satellites are separated from the main peak Fe2p_3/2_ at 5.7, 8.5, and 8.8 eV. The high value of the binding energy and the presence of core-level satellites allow us to state that iron in the oxidized catalysts is in the Fe^3+^ state.

It is known that one of the reasons for catalyst deactivation is the formation of polymerized products on the catalyst surface. These products are formed due to both the self-condensation of furfural molecules and the interaction of furfural with FA [18]. The formation of these products is an undesirable process because they reduce the activity of the catalyst and are coke precursors, carburizing the surface and blocking the access of the substrate to the catalyst’s active sites. To determine the carbon amount in the catalysts after the reaction, an elemental CHNS analysis was used. The results of this analysis are shown in Table 5. The carbon content in monometallic catalysts is approximately 3–8 wt.%, while in the case of the most active catalysts (15Cu-SiO_2_ and 40Cu-SiO_2_), the carbon content does not exceed 3.5 wt.%. The introduction of iron into the 15Cu-SiO_2_ catalyst increases the formation of carbon deposits, which block the active centers of the catalysts, thus causing their lower activity compared to the pristine sample. The introduction of iron and aluminum into the 40Cu-SiO_2_ catalyst has no substantial effect on the carbon content in the samples after the reaction. It can be assumed that iron and aluminum oxides can be covered with carbon deposits while Cu-containing particles remain free, thus providing the high activity of those catalysts.

To confirm this assumption, the structural features and morphology of the most active 35Cu13Fe1Al-SiO_2_ catalyst (fresh and after reaction) were studied by high-resolution transmission electron microscopy (HRTEM). The fresh 35Cu13Fe1Al-SiO_2_ catalyst reduced in a tubular reactor at 200 °C and passivated by ethanol was studied by HRTEM mapping. Analysis using this method shows that the catalyst has a matrix of amorphous SiO_2_ (Appendix A), in which Fe_2_O_3_ clusters (presumably), less than 1 nm in size, are distributed (Appendix A and Figure 9a). Aluminum oxide is also a finely dispersed phase; it is not possible to determine more accurately due to the small particle size and low concentration. In the “cavities” of matrix space is copper. Copper is represented by several sizes and phases (Figure 9b): small particles (10–50 nm) of metallic copper, covered with an oxide film due to passivation with ethanol after reduction, and large particles up to 100 nm.

The morphology of the catalyst after the reaction does not significantly change; there is no change in the size of Cu-containing particles, which indicates a stabilization effect due to the modification with iron and aluminum (Figure 10a). However, the size of iron-containing particles increases to 2 nm after the reaction, and it is possible to determine that iron is a phase of hematite Fe_2_O_3_ (Figure 10b). The amount of formed carbon is 3–4 wt.%, which correlates with the data of the CHNS analysis (Table 5). On the HRTEM images, carbon is identified on the surface of the copper-containing particles, and the coating thickness corresponds to 1–2 monolayers of carbon (Figure 10c).

It is important to note that the distribution of Fe, Al, and C elements in the patterns of 35Cu13Fe1Al-SiO_2_ are similar (Figure 11). It confirms that carbon deposits are formed on iron- and aluminum-containing particles.

Based on the data of physicochemical methods, the proposed 35Cu13Fe1Al-SiO_2_ catalyst represents an amorphous SiO_2_ matrix with a high surface area in which nanosized hematite particles are uniformly distributed. The active component of the catalyst is presented in the form of metallic copper of various sizes (small particles of 10–50 nm and large particles up to 100 nm), which is covered with copper oxides. After the reaction, carbon covers copper particles with a thickness of 1–2 monolayers. It can also be argued that iron and aluminum oxides are strongly covered with carbon deposits, while Cu-containing particles remain relatively free, which can provide the high activity of that catalyst.

## 3. Materials and Methods

### 3.1. Chemicals

All chemical reagents used in this study were commercially available, with no additional purification applied. Nickel(II) nitrate hexahydrate Ni(NO_3_)_2_·6H_2_O (≥98%, Reakhim JSC, Moscow, Russia), cupric (II) carbonate basic CuCO_3_·Cu(OH)_2_ (≥96%, Reakhim JSC, Moscow, Russia), ammonia solution NH_4_OH (25%, LenReaktiv JSC, Sankt-Peterburg, Russia), iron nitrate nonahydrate Fe(NO_3_)_3_·9H_2_O (≥98%, Soyuzkhimprom JSC, Novosibirsk, Russia), aluminum nitrate nonahydrate Al(NO_3_)_3_·9H_2_O (≥98%, Reakhim JSC, Moscow, Russia), and ethyl silicate (≥99%, Reakhim JSC, Moscow, Russia) were used for the catalyst preparation. Isopropyl alcohol (≥99.8%, Reakhim JSC, Moscow, Russia), furfural (≥99.5%, Component-Reactiv LLC, Moscow, Russia), argon (≥99.99%, Pure Gases Ltd., Novosibirsk, Russia), and hydrogen (≥99.99%, Pure Gases Ltd., Novosibirsk, Russia) were used in the catalytic tests. Furfuryl alcohol (≥98%, Component Reactiv JSC, Moscow, Russia), 2-methylfuran (99%, Acros Organics, NJ, USA), and 2-methyltetrahydrofuran (≥99%, Acros Organics, NJ, USA) were used to determine the relative response factors.

### 3.2. Catalyst Preparation

Two approaches based on the sol–gel method (i.e., homophase and heterophase sol–gel methods) were used for the catalyst preparation [48,49]. The homophase sol–gel method was used for the synthesis of samples with copper content up to 30 wt.%. In the case of CuFe-containing catalysts, the required amounts of copper nitrate and iron nitrate were dissolved in 150 mL of distilled water in a glass beaker for the preparation of samples by the homophase sol–gel method (HomSG series). The beaker was mounted on a stand with a tripod, on which a top-drive stirrer with a glass/PTFE propeller stirrer was fixed. Stirring was carried out until the complete dissolution of salts. Then, a pH-meter electrode was immersed in the prepared solution of the precursors, and 75 mL of ethyl silicate (ES) was added under continuous stirring at 700 rpm. Nitric acid was added dropwise until pH ≈ 1.3–1.5 and stirring was continued for 40 min to hydrolyze ES (step of sol formation). Then, a solution of ammonia was added dropwise until pH = 7. Upon reaching pH ≈ 5–6, the stirring speed was increased up to 14,000–17,000 rpm due to the formation of a viscous pasty mixture (step of gel formation). The resulting precursor was placed in an evaporating bowl and left in a muffle furnace at a temperature of 160 °C overnight. The dried samples were calcined in the following mode: 200 °C for 30 min → 250 °C for 30 min → 300 °C for 30 min → 350 °C for 30 min → 400 °C for 2 h. The prepared catalysts in an oxide form are referred to as xCuyFe-SiO_2_, where x and y are the mass contents of Cu and Fe in metal form, respectively (see Table 2).

To obtain the catalysts with a high Cu loading (40 and 50 wt.%), the heterophase sol–gel method (HetSG series) was used. The required amounts of copper carbonate, iron nitrate (in the case of CuFe-containing catalysts), and aluminum nitrate (in the case of CuFeAl-containing catalyst) were dissolved in 95 mL of distilled water, placed into a 250 mL low beaker, and intensively mixed at 14,000–17,000 rpm using a top-drive stirrer. Ammonia solution was added dropwise to the resulting suspension until pH = 10. Then, the required amount of ES was added and stirred at 14,000–17,000 rpm for 15 min. The obtained precursor was placed in an evaporating bowl and left in a muffle furnace at a temperature of 160 °C overnight. The calcination was carried out in the following mode: 200 °C for 30 min → 250 °C for 30 min → 300 °C for 30 min → 350 °C for 30 min → 400 °C for 2 h. The prepared catalysts in oxide form are referred to as xCuyFezAl-SiO_2_, where x, y, and z are the mass contents of Cu, Fe, and Al in metal form, respectively (see Table 2).

### 3.3. Catalyst Activity Tests

Activity and selectivity of the prepared catalysts were studied in the process of furfural hydro conversion. The experiments were carried out in a 300 mL batch reactor (autoclave) equipped with a mechanical stirrer, an electric furnace, and an operating system for controlling external and internal temperatures, pressure, and stirring rate of the reaction mixture. Before the reaction, 0.3 or 1 g of powdered catalyst was placed in the autoclave and activated in a hydrogen flow (300 mL/min) at a temperature of 200 °C for 1 h. After cooling to room temperature, 60 mL of a solution of 7 vol.% furfural in isopropanol was added, which corresponds to the furfural/catalyst mass ratio of 16.2 or 4.9%, depending on the initial mass of the catalyst. Next, the reactor was sealed, and the mixture was heated to the required temperature (100–250 °C) with a heat rate of 10 °C/min. After reaching the target temperature, hydrogen was supplied to the reactor until a pressure of 5.0 MPa, and the mixture was intensively stirred at 1800 rpm. The start and end times of the reaction corresponded to the moment when the stirring was turned on and off, respectively.

### 3.4. Product Analysis

Qualitative analysis of liquid reaction products was carried out on an Agilent 7000B GC/MS (Agilent Technologies Inc., Santa Clara, CA, USA) with a triple quadrupole analyzer and an HP-5 MS quartz capillary column ((5%-phenyl)-methylpolysiloxane, length of 30 m, inner diameter of 0.25 mm, phase thickness of 0.25 μm) from Agilent Technologies Inc (Santa Clara, CA, USA). The temperature program was as follows: 50 °C/min for 3 min, then 10 °C/min to 260 °C/min. Mass spectra recording conditions: electron ionization (70 eV), scanning mode in the m/z range of 40–500. Helium was used as a carrier gas. The NIST.11. database was used to identify the components of the analyzed sample.

Quantitative analysis of liquid reaction products was carried out using an Agilent GC-7820A gas chromatograph (Agilent Technologies Inc., Santa Clara, CA, USA) equipped with a flame ionization detector and CM-Wax column (stationary phase polyethylene glycol, 30 m×0,25 mm, phase thickness 0.25 μm) from JSC CXM (Moscow, Russia) and HP-5 capillary column (stationary phase ((5%-phenyl)-methylpolysiloxane, length of 30 m, inner diameter of 0.32 mm, phase thickness of 0.25 μm) from Agilent Technologies Inc. (Santa Clara, CA, USA). Argon was used as a carrier gas with a flow rate of 25 mL/min. The temperature program was as follows: 4 min—constant temperature for 50 °C, then heat at 8 °C/min to 62 °C, 12 °C/min to 146 °C, 20 °C/min to 190 °C and 10 °C/min to 230 °C. The analysis time was 19.2 min. The samples were injected in an amount of 0.1 µL using a chromatographic syringe. The product quantification was determined using the normalization method with the relative response factors of 1.03, 1.00, 1.25, and 1.44 for furfural, furfuryl alcohol, 2-methylfuran, and 2-methyltetrahydrofuran, respectively. The relative yield of the reaction products (%) was estimated as the molar ratio of the amount of formed product to the initial amount of furfural multiplied by 100%. The components of the reaction mixture were identified by retention times. The accuracy of the chromatographic analysis was 5%.

### 3.5. Elemental Composition of Catalysts

Elemental analysis of the fresh catalysts in oxide form was carried out using atomic emission spectroscopy with inductively coupled plasma (ICP–AES) on an Optima 4300 DV (Perkin Elmer Inc., Waltham, MA, USA).

### 3.6. Texture Characteristics

The texture properties of catalysts were analyzed by low-temperature nitrogen porosimetry using an automated volumetric adsorption station ASAP-2400 (Micromeritics Instrument Corp., Norcross, GA, USA). The Brunauer–Emmett–Teller method was used for data processing. Before nitrogen adsorption, the samples were degassed for 4 h at 150 °C and a pressure of 0.13 Pa.

### 3.7. CO Chemosorption

CO pulse chemisorption measurements using a Chemosorb analyzer (JSC SLO, Moscow, Russia) were used to estimate the number of active sites in the synthesized catalysts according to the previously published technique [62]. Before analysis, the catalysts were reduced at 200 °C in a H_2_ flow. The samples were cooled to 25 °C in an Ar atmosphere, and CO was pulsed into the reactor until complete sample saturation was observed. The total CO uptake was used to calculate the number of active sites.

### 3.8. High-Resolution Transmission Electron Microscopy

The structure and microstructure of the catalysts were studied by high-resolution transmission electron microscopy (HRTEM) using a ThemisZ electron microscope (Thermo Fisher Scientific, Waltham, MA, USA) with an accelerating voltage of 200 kV and a limiting resolution of 0.07 nm. Images were recorded using a Ceta 16 CCD array (Thermo Fisher Scientific, Waltham, MA, USA). The instrument is equipped with a SuperX (Thermo Fisher Scientific, Waltham, MA, USA) energy-dispersive characteristic X-ray spectrometer (EDX) with a semiconductor Si detector with an energy resolution of 128 eV. For electron microscopy studies, sample particles were deposited on perforated carbon substrates fixed on copper or molybdenum grids using a UZD-1UCH2 ultrasonic disperser. The sample was suspended in an alcohol solution and placed on an ultrasonic disperser. Ultrasonic treatment resulted in the evaporation of liquid and deposition of sample particles on a copper mesh.

### 3.9. X-ray Diffraction

The phase composition of the catalysts in the initial oxide state and after the reaction was studied by X-ray diffraction (XRD). An investigation was performed on a Thermo X’tra diffractometer (Bruker, Billerica, MA, USA) in the angle range of 10–72° with a step of 2θ = 0.02° and a speed of 2°/min with a Mythen2R 1D linear detector (Decstris, Baden, Switzerland) and using monochromatized CuKα radiation (λ = 1.5418 Å). The average sizes of the coherent scattering regions were calculated using the Scherrer formula for the most intense reflections. The refinement of the lattice parameters and phase ratios was carried out by the Rietveld method.

### 3.10. X-ray Photoelectron Spectroscopy (XPS)

The chemical composition of the catalyst surface was studied by X-ray photoelectron spectrometry (SPECS Surface Nano Analysis GmbH, Berlin, Germany). The spectrometer was equipped with a hemispherical analyzer PHOIBOS-150-MCD-9 and an XR-50 source of X-ray characteristic radiation with a double Al/Mg anode. The spectra were recorded using nonmonochromatized AlKα radiation (1486.61 eV). The binding energy scale (E_b_) was calibrated by the internal standard method using the Si2p line of silicon included in the support (E_b_ = 103.3 eV). The relative concentrations of elements were determined by integral intensities of XPS lines, considering the photoionization cross-section of the corresponding terms [63]. The spectra were decomposed into individual components for a detailed analysis. After subtracting the background by the Shirley method, the experimental curve was decomposed into a series of lines corresponding to the photoemission of electrons from atoms in various chemical surroundings. The XPS data were processed using the CasaXPS 2.3.25 software package [64]. The shape of the peaks was approximated by a symmetric function obtained by convolution of the Gauss and Lorentzian functions.

### 3.11. Determination of Carbon Content

The carbon content in the catalysts after the reaction was determined using a Vario El Cube elemental (CHNS/O) analyzer (Elementar Analysensysteme GmbH, Langenselbold, Germany) equipped with a high-temperature combustion unit and a thermal conductivity detector. The details on CHNS analysis in solid samples can be found elsewhere [65].

## 4. Conclusions

A new type of Cu-containing catalysts prepared by homo- and heterophase sol–gel methods using iron and aluminum modifiers has been proposed. The used preparation techniques make it possible to obtain the catalyst with a highly dispersed active component, even at a high Cu content. The synthesized catalysts have a high surface area and exhibit high activity in the hydroconversion of furfural selectively to FA or 2-MF, depending on the reaction temperature.

The active component in these catalysts is copper-containing particles of different sizes, namely, small particles of 10–50 nm and larger particles in size up to 100 nm, which are distributed in the SiO_2_ matrix. The incorporation of iron and aluminum in Cu-containing particles leads to a decrease in the particle size of the active component and, consequently, an increase in the activity compared to the activity of the corresponding monometallic catalyst. Modification with iron also increases the surface area of the catalyst by four times and prevents the agglomeration of copper particles during the reaction. In the synthesized catalysts, iron presents in the form of hematite nanoparticles (1–2 nm), while aluminum forms nanoparticles of aluminum oxide.

Considering all studied catalysts, the 35Cu13Fe1Al-SiO_2_ sample exhibits the highest activity and selectivity in the formation of FA from furfural in a batch reactor. The 100% conversion of furfural with 97–99% yield of FA is achieved for this catalyst at 100–140 °C, H_2_ pressure of 5.0 MPa, catalyst loading of 0.3–1.0 g, and the reaction time of 2.5–5.0 h. At 250 °C, the main product is 2-MF with selectivity up to 76% at 100% conversion of furfural. It is shown that hematite and aluminum oxide activate the C=O bonds of furfural. Additionally, iron and aluminum oxides are covered with carbon deposits during the reaction, while the surface of Cu-containing particles remains relatively free, thus providing the high activity of the 35Cu13Fe1Al-SiO_2_ catalyst.

## Figures and Tables

**Figure 1 ijms-24-07547-f001:**
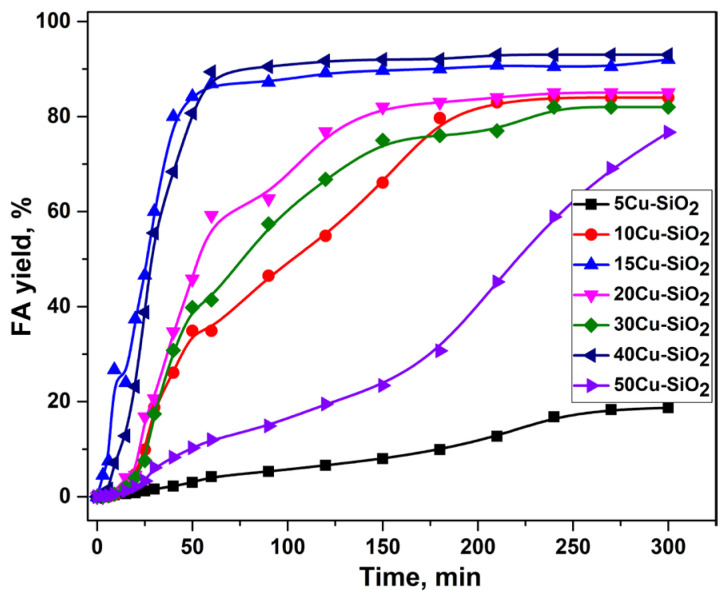
Dependence of FA yield on the reaction time in the presence of monometallic copper catalysts in a batch reactor. Reaction conditions: P(H_2_) = 5.0 MPa, 100 °C, 7 vol.% of furfural in isopropyl alcohol, 4.5 h, m_cat_ = 1 g.

**Figure 2 ijms-24-07547-f002:**
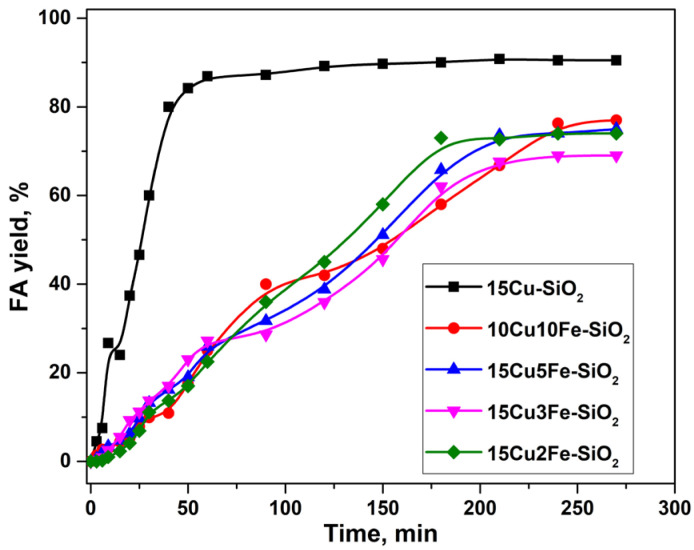
Dependence of FA yield on the reaction time in the presence of CuFe-containing catalysts (HomSG series) in a batch reactor. Reaction conditions: P(H_2_) = 5.0 MPa, 100 °C, 7 vol.% of furfural in isopropyl alcohol, 4.5 h, m_cat_ = 1 g.

**Figure 3 ijms-24-07547-f003:**
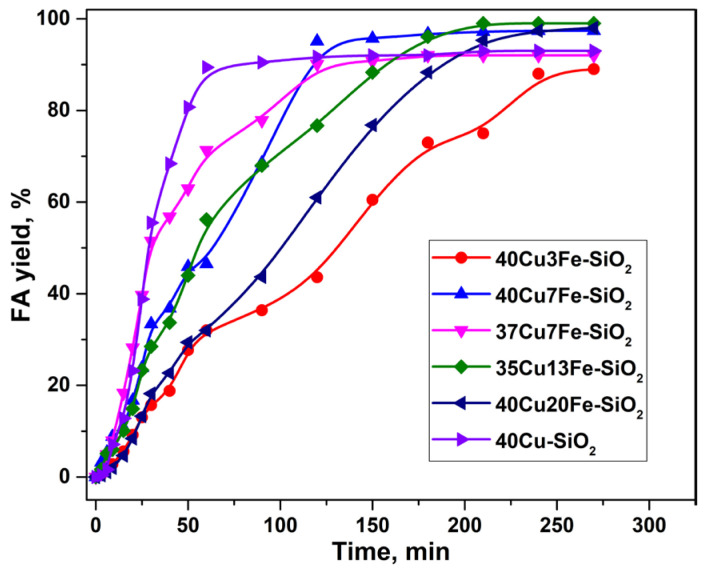
Dependence of FA yield on the reaction time in the presence of CuFe-containing catalysts (HeteroSG series) in a batch reactor. Reaction conditions: P(H_2_) = 5.0 MPa, 100 °C, 7 vol.% of furfural in isopropyl alcohol, 4.5 h, m_cat_ = 1 g.

**Figure 4 ijms-24-07547-f004:**
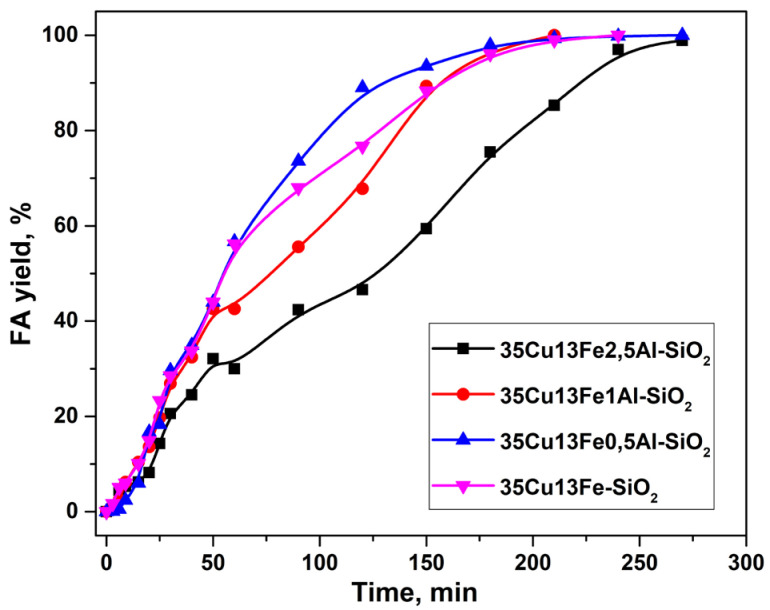
Dependence of FA yield on the reaction time in the presence of CuFeAl-containing catalysts (HeteroSG series) in a batch reactor. Reaction conditions: P(H_2_) 5.0 MPa, 100 °C, 7 vol.% of furfural in isopropyl alcohol, 4.5 h, m_cat_ = 1 g.

**Figure 5 ijms-24-07547-f005:**
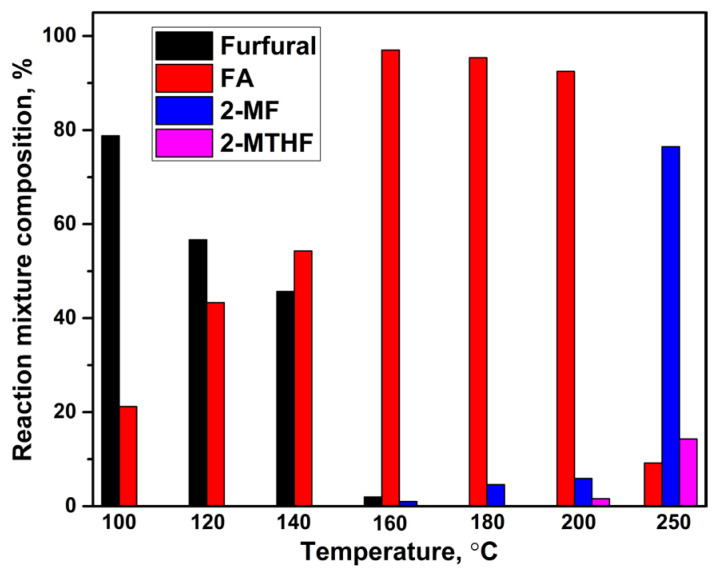
Dependence of reaction mixture composition on temperature for the 35Cu13Fe1Al-SiO_2_ catalyst in a batch reactor. Reaction conditions: P(H_2_) = 5.0 MPa, 100 °C, 7 vol.% of furfural in isopropyl alcohol, 2 h, m_cat_ = 0.3 g. FA is furfuryl alcohol, 2-MF is 2-methylfuran, 2-MTHF is 2-methyltetrahydrofuran.

**Figure 6 ijms-24-07547-f006:**
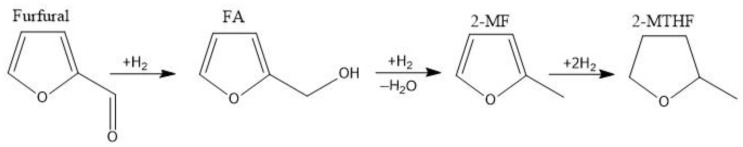
Schema of furfural hydroconversion in the presence of 35Cu13Fe1Al-SiO_2_ catalyst in a batch reactor (FA is furfuryl alcohol, 2-MF is 2-methylfuran, 2-MTHF is 2-methyltetrahydrofuran).

**Figure 7 ijms-24-07547-f007:**
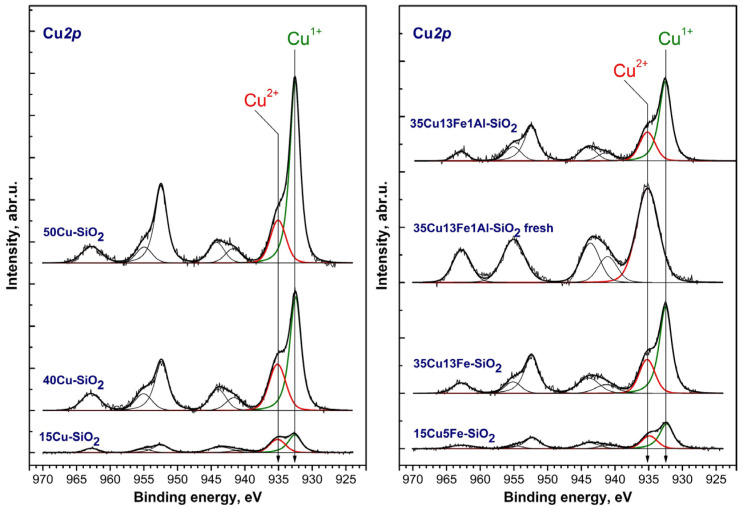
Cu2p spectra of Cu-containing catalysts (the spectra are normalized to the integrated intensity of the corresponding Si2p spectra).

**Figure 8 ijms-24-07547-f008:**
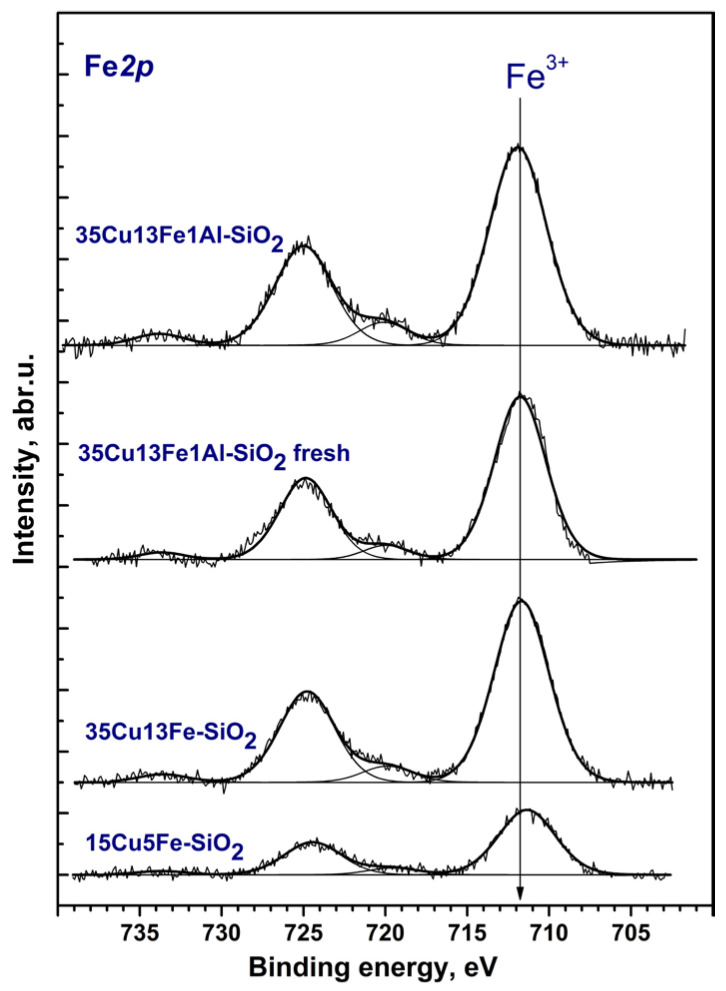
Fe2p spectra of the Cu-containing catalysts (the spectra are normalized to the integrated intensity of the corresponding Si2p spectra).

**Figure 9 ijms-24-07547-f009:**
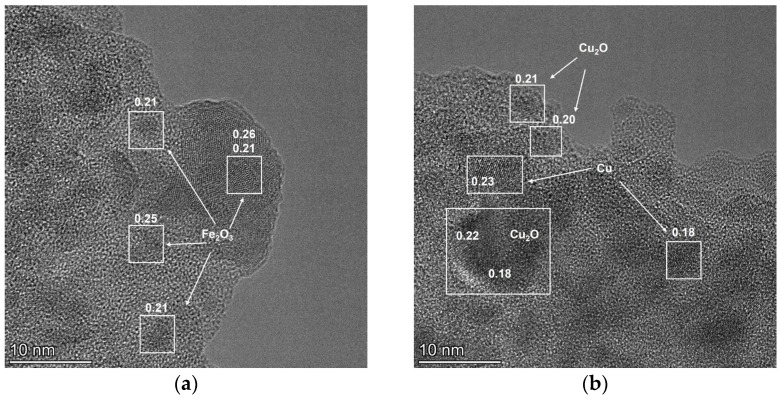
HRTEM images of reduced and passivated 35Cu13Fe1Al-SiO_2_ catalyst: Fe_2_O_3_ particles (**a**), Cu and Cu_2_O particles (**b**). The numbers indicate the interplanar distances of the indicated phases (nm).

**Figure 10 ijms-24-07547-f010:**
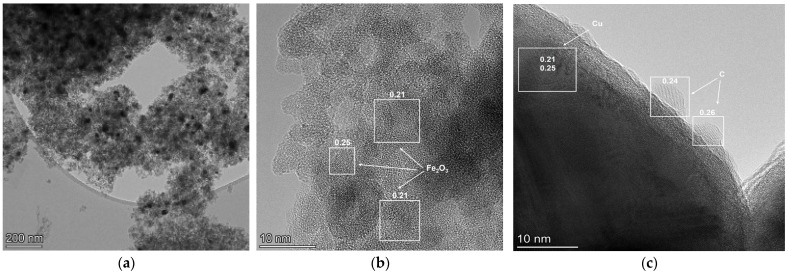
HRTEM images of 35Cu13Fe1Al-SiO_2_ catalyst after reaction: perspective view (**a**), Fe_2_O_3_ particles (**b**), carbon on the surface of copper-containing particles (**c**). The numbers indicate the interplanar distances of the indicated phases (nm).

**Figure 11 ijms-24-07547-f011:**
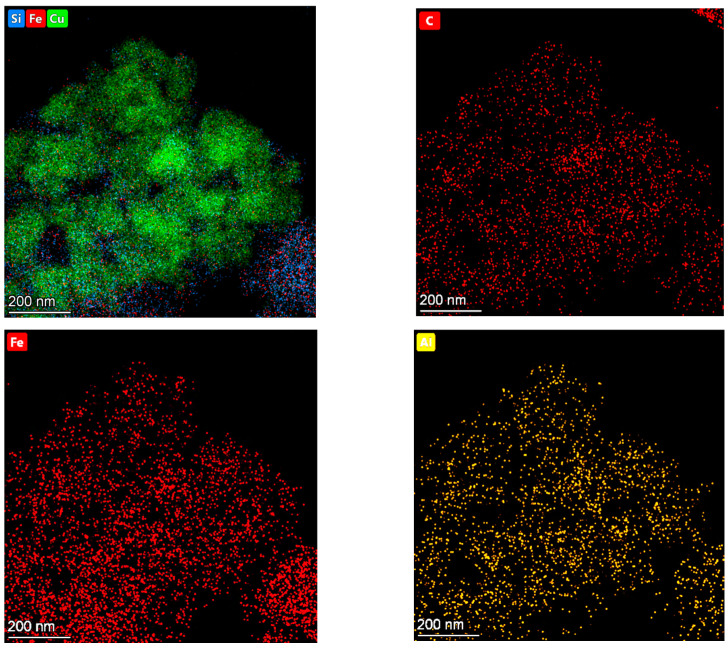
EDS elemental mapping images of 35Cu13Fe1Al-SiO_2_ catalyst after reaction.

**Table 1 ijms-24-07547-t001:** Correlation between the amount of CO chemisorbed by the reduced monometallic xCu–SiO_2_ catalyst and the FA yield. Reaction conditions: P(H_2_) = 5.0 MPa, 100 °C, 7 vol.% of furfural in isopropyl alcohol, reaction time is 5 h, m_cat_ = 1 g.

Catalyst	CO Uptake, µmol/g_cat._	Yield of FA, %
HomSG series ^a^		
5Cu-SiO_2_	26	19
10Cu-SiO_2_	65	84
15Cu-SiO_2_	89	92
20Cu-SiO_2_	80	85
30Cu-SiO_2_	65	82
HetSG series ^b^		
40Cu-SiO_2_	86	93
50Cu-SiO_2_	45	77

^a^ A series of catalysts obtained via the homophase sol–gel method. ^b^ A series of catalysts obtained via the heterophase sol–gel method.

**Table 2 ijms-24-07547-t002:** Textural characteristics and phase composition of Cu-containing catalysts.

Catalyst ^a,b^	SSA ^c^, m^2^/g	V_∑_, cm^3^/g	Phase Composition/CSR, Å ^d^
HomSG series ^e^			
5Cu-SiO_2_	156	0.71	SiO_2_ (amorph. ^f^)/- ^g^
10Cu-SiO_2_	210	0.73	SiO_2_ (amorph.)/- CuO/40
15Cu-SiO_2_	290	0.78	SiO_2_ (amorph.)/- CuO/40
20Cu-SiO_2_	320	0.78	SiO_2_ (amorph.)/- CuO/50
30Cu-SiO_2_	329	0.82	SiO_2_ (amorph.)/- CuO/90
10Cu10Fe-SiO_2_ (0.9) ^h^	215	1.05	SiO_2_ (amorph.)/-
15Cu5Fe-SiO_2_ (2.4)	257	1.03	SiO_2_ (amorph.)/-
15Cu3Fe-SiO_2_ (4.4)	262	1.00	SiO_2_ (amorph.)/-
15Cu2Fe-SiO_2_ (5.5)	270	0.99	SiO_2_ (amorph.)/-
HetSG series ^i^			
40Cu-SiO_2_	83	0.45	SiO_2_ (amorph.)/- CuO/130
50Cu-SiO_2_	40	0.20	CuO/170
40Cu20Fe-SiO_2_ (1.8)	360	0.72	SiO_2_ (amorph.)/- CuO (trace)
35Cu13Fe-SiO_2_ (2.4)	348	0.73	SiO_2_ (amorph.)/- CuO (trace)/-
37Cu7Fe-SiO_2_ (4.4)	330	0.72	SiO_2_ (amorph.)/- CuO (trace)/-
40Cu7Fe-SiO_2_ (5.5)	323	0.72	SiO_2_ (amorph.)/- CuO (trace)/-
40Cu3Fe-SiO_2_ (11.6)	310	0.72	SiO_2_ (amorph.)/- CuO (trace)/-
35Cu13Fe2.5Al-SiO_2_ (1) ^j^	235	0.75	CuO/160
35Cu13Fe1Al-SiO_2_ (2)	240	0.78	CuO/170
35Cu13Fe0.5Al-SiO_2_ (4.6)	260	0.82	CuO/320

^a^ Numbers in the catalyst notation correspond to the percentage of Cu, Fe, and Al in the metal form (wt.%). ^b^ The catalyst composition was determined by atomic emission spectroscopy with inductively coupled plasma (ICP–AES). ^c^ Specific surface area was determined via the Brunauer–Emmett–Teller method, using nitrogen adsorption isotherms. ^d^ Determined by the X-ray diffraction method (conditions are described below in Section 3.9). ^e^ Catalyst series obtained via the homophase sol–gel method. ^f^ Amorphous. ^g^ Not determined due to X-ray amorphous. ^h^ Cu/Fe molar ratio. ^i^ Catalyst series obtained via the heterophase sol–gel method. ^j^ Fe/Al molar ratio.

**Table 3 ijms-24-07547-t003:** Phase composition and corresponding CSR sizes for the catalysts after reaction.

Catalyst after Reaction	Phase Composition, %	CSR, Å
40Cu-SiO_2_	30% Cu_2_O 70% Cu	30 140
35Cu13Fe-SiO_2_	50% Cu_2_O 50% Cu	50 140
35Cu13Fe1Al-SiO_2_	25% Cu_2_O 75% Cu	55 240

**Table 4 ijms-24-07547-t004:** Atomic ratios of elements in the surface layer of the catalysts.

Catalyst	[Cu]/[Si]	[Fe]/[Si]	[O]/[Si]	[C]/[Si]	%, Cu^2+^	%, Cu^1+^	%, Cu^0^
15Cu-SiO_2_	0.082	0.000	2.27	0.51	55	45	0
40Cu-SiO_2_	0.378	0.000	2.33	1.75	47	53	0
50Cu-SiO_2_	0.519	0.000	2.40	2.32	36	64	0
15Cu5Fe-SiO_2_	0.074	0.025	2.32	0.77	47	53	0
35Cu13Fe-SiO_2_	0.224	0.068	2.44	0.87	43	57	0
35Cu13Fe1Al-SiO_2_ (fresh)	0.336	0.053	2.27	0	100	0	0
35Cu13Fe1Al-SiO_2_	0.195	0.073	2.42	0.85	42	58	0

**Table 5 ijms-24-07547-t005:** Carbon content in the catalysts after reaction determined by CHNS analysis.

Catalyst	Carbon Content, wt.%
5Cu-SiO_2_	7.6 ± 0.5
10Cu-SiO_2_	4.5 ± 0.4
15Cu-SiO_2_	2.80 ± 0.05
20Cu-SiO_2_	5.68 ± 0.08
30Cu-SiO_2_	3.16 ± 0.02
40Cu-SiO_2_	3.0 ± 0.1
50Cu-SiO_2_	6.3 ± 0.6
10Cu10Fe-SiO_2_	8.8 ± 0.3
15Cu5Fe-SiO_2_ (2.4)	5.8 ± 0.1
15Cu3Fe-SiO_2_ (4.4)	6.82 ± 0.10
15Cu2Fe-SiO_2_ (5.5)	6.2 ± 0.8
35Cu13Fe-SiO_2_	4.0 ± 0.2
40Cu20Fe-SiO_2_	4.33 ± 0.03
37Cu7Fe-SiO_2_	4.72 ± 0.03
40Cu7Fe-SiO_2_	3.9 ± 0.1
40Cu3Fe-SiO_2_	4.8 ± 0.5
35Cu13Fe0,5Al-SiO_2_	4.2 ± 0.6
35Cu13Fe1Al-SiO_2_	3.5 ± 0.2
35Cu13Fe2.5Al-SiO_2_	4.3 ± 0.5

## Data Availability

Not applicable.

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
