# Peer review of "High-Loaded Copper-Containing Sol–Gel Catalysts for Furfural Hydroconversion"

_ijms, 2023, doi:10.3390/ijms24087547_

Round 1
Reviewer 1 Report
In this paper, a new type of copper catalyst was prepared by homogeneous and heterogeneous sol-gel method, and the catalyst was modified with iron and aluminum, which was used in the hydrogenation of furfural to furfural and 2-methylfuran. It is a challenging subject, and the research in this paper has substantial significance. Publication may be considered after some minor issues are involved. The following minor issues need to be resolved:
1. For example, the subscripts of "a" and "b" in Table 1 are the same.
2. Please carefully check the language and format of the manuscript. For example, the subscript of "SiO2" in Table 5, the fourth line of section 3.1, "≥25%, %, JSC", etc.
3. The discussion and background should be broader and more in-depth, the elaboration should be more detailed, and more relevant references should be added appropriately.
4. The examples listed in this paper are few and the discussion is not complete enough. Appropriate examples can be given and more literatures can be referred to.
Author Response
Reviewer #1
In this paper, a new type of copper catalyst was prepared by homogeneous and heterogeneous sol-gel method, and the catalyst was modified with iron and aluminum, which was used in the hydrogenation of furfural to furfural and 2-methylfuran. It is a challenging subject, and the research in this paper has substantial significance. Publication may be considered after some minor issues are involved. The following minor issues need to be resolved:
Response
Thank you for your interest of our work and your valuable comments, helping us to improve this paper. A detailed response to each comment is provided below.
- For example, the subscripts of "a" and "b" in Table 1 are the same.
Response
We have corrected the subscripts of "a" and "b" in Table 1. (a Catalyst series obtained via the homophase sol-gel method. b Catalyst series obtained via the heterophase sol-gel method.). Please find the revised version of manuscript.
- Please carefully check the language and format of the manuscript. For example, the subscript of "SiO2" in Table 5, the fourth line of section 3.1, "≥25%, %, JSC", etc.
Response
We have checked the language and format of the manuscript. The subscripts in Table 5 and typos in the manuscript are eliminated. Please find the revised version of manuscript.
According to your recommendation, English language in the manuscript was thoroughly checked and edited. The revised version of manuscript was checked using AJE Grammar Check tool (https://www.aje.com/grammar-check/). This tool shows 7.8/10 score (90th percentile of papers submitted to AJE) and indicates that the manuscript is well written and does not need language editing.
- The discussion and background should be broader and more in-depth, the elaboration should be more detailed, and more relevant references should be added appropriately.
Response
According to your recommendation, we have added to Introduction some information on application of furfuryl alcohol and 2-methylfuran. Also, we have cited some references concerning the deactivation of Cu-Cr catalysts (e.g., coating of copper by Cr-containing particles). Additional information on the quantitative determination of products by gas chromatography and the methodology of catalytic tests has added (Section 3.3 and 3.4). Please find the revised version of manuscript.
- Yan, K.; Chen, A. Efficient Hydrogenation of Biomass-Derived Furfural and Levulinic Acid on the Facilely Synthesized Noble-Metal-Free Cu–Cr Catalyst. Energy 2013, 58, 357–363, doi:10.1016/j.energy.2013.05.035.
- Sulmonetti, T.P.; Hu, B.; Ifkovits, Z.; Lee, S.; Agrawal, P.K.; Jones, C.W. Vapor Phase Hydrogenolysis of Furanics Utilizing Reduced Cobalt Mixed Metal Oxide Catalysts. ChemCatChem 2017, 9, 1815–1823, doi:10.1002/cctc.201700228.
- Date, N.S.; Hengne, A.M.; Huang, K.-W.; Chikate, R.C.; Rode, C.V. Single Pot Selective Hydrogenation of Furfural to 2-Methylfuran over Carbon Supported Iridium Catalysts. Green Chem. 2018, 20, 2027–2037, doi:10.1039/C8GC00284C.
- Varila, T.; Mäkelä, E.; Kupila, R.; Romar, H.; Hu, T.; Karinen, R.; Puurunen, R.L.; Lassi, U. Conversion of Furfural to 2-Methylfuran over CuNi Catalysts Supported on Biobased Carbon Foams. Catalysis Today 2021, 367, 16–27, doi:10.1016/j.cattod.2020.10.027.
- Corma, A.; de la Torre, O.; Renz, M.; Villandier, N. Production of High-Quality Diesel from Biomass Waste Products. Angewandte Chemie International Edition 2011, 50, 2375–2378, doi:10.1002/anie.201007508.
- Corma, A.; Torre, O. de la; Renz, M. Production of High Quality Diesel from Cellulose and Hemicellulose by the Sylvan Process: Catalysts and Process Variables. Energy Environ. Sci. 2012, 5, 6328–6344, doi:10.1039/C2EE02778J.
- Zeitsch, K.J. The Chemistry and Technology of Furfural and Its Many By-Products; Elsevier, 2000; ISBN 978-0-08-052899-1.
- Taylor, M.J.; Durndell, L.J.; Isaacs, M.A.; Parlett, C.M.A.; Wilson, K.; Lee, A.F.; Kyriakou, G. Highly Selective Hydrogenation of Furfural over Supported Pt Nanoparticles under Mild Conditions. Applied Catalysis B: Environmental 2016, 180, 580–585, doi:10.1016/j.apcatb.2015.07.006.
- Yan, K.; Wu, G.; Lafleur, T.; Jarvis, C. Production, Properties and Catalytic Hydrogenation of Furfural to Fuel Additives and Value-Added Chemicals. Renewable and Sustainable Energy Reviews 2014, 38, 663–676.
- Wang, Y.; Zhao, D.; Rodríguez-Padrón, D.; Len, C. Recent Advances in Catalytic Hydrogenation of Furfural. Catalysts 2019, 9, 796, doi:10.3390/catal9100796.
- Schneider, M.H.; Phillips, J.G. Furfuryl Alcohol and Lignin Adhesive Composition 2004.
12.Barr, J.B.; Wallon, S.B. The Chemistry of Furfuryl Alcohol Resins. Journal of Applied Polymer Science 1971, 15, 1079–1090, doi:10.1002/app.1971.070150504.
- Recent Advances in the Catalytic Transfer Hydrogenation of Furfural to Furfuryl Alcohol over Heterogeneous Catalysts. Green Chemistry 2022, 24, 1780–1808, doi:10.1039/d1gc04440k.
- Vaidya, P.D.; Mahajani, V.V. Kinetics of Liquid-Phase Hydrogenation of Furfuraldehyde to Furfuryl Alcohol over a Pt/C Catalyst. Ind. Eng. Chem. Res. 2003, 42, 3881–3885, doi:10.1021/ie030055k.
- Sitthisa, S.; Sooknoi, T.; Ma, Y.; Balbuena, P.B.; Resasco, D.E. Kinetics and Mechanism of Hydrogenation of Furfural on Cu/SiO2 Catalysts. Journal of Catalysis 2011, 277.
- Zhang, H.; Lei, Y.; Kropf, A.J.; Zhang, G.; Elam, J.W.; Miller, J.T.; Sollberger, F.; Ribeiro, F.; Akatay, M.C.; Stach, E.A.; et al. Enhancing the Stability of Copper Chromite Catalysts for the Selective Hydrogenation of Furfural Using ALD Overcoating. Journal of Catalysis 2014, 317, 284–292, doi:10.1016/j.jcat.2014.07.007.
- Liu, D.; Zemlyanov, D.; Wu, T.; Lobo-Lapidus, R.J.; Dumesic, J.A.; Miller, J.T.; Marshall, C.L. Deactivation Mechanistic Studies of Copper Chromite Catalyst for Selective Hydrogenation of 2-Furfuraldehyde. Journal of Catalysis 2013, 299, 336–345, doi:10.1016/j.jcat.2012.10.026.
- The examples listed in this paper are few and the discussion is not complete enough. Appropriate examples can be given and more literatures can be referred to.
Response
According to your recommendation, we have provided some references concerning chemisorption of CO on the Cu-containing catalysts. Please find the revised version of manuscript.
- Parris, G.E.; Klier, K. The Specific Copper Surface Areas in CuZnO Methanol Synthesis Catalysts by Oxygen and Carbon Monoxide Chemisorption: Evidence for Irreversible CO Chemisorption Induced by the Interaction of the Catalyst Components. Journal of Catalysis 1986, 97, 374–384, doi:10.1016/0021-9517(86)90009-6.
- Phillips, J.M.; Leibsle, F.M.; Holder, A.J.; Keith, T. A Comparative Study of Chemisorption by Density Functional Theory, Ab Initio, and Semiempirical Methods: Carbon Monoxide, Formate, and Acetate on Cu(110). Surface Science 2003, 545, 1–7, doi:10.1016/j.susc.2003.07.006.
- Smirnov, A.A.; Khromova, S.A.; Bulavchenko, O.A.; Kaichev, V.V.; Saraev, A.A.; Reshetnikov, S.I.; Bykova, M.V.; Trusov, L.I.; Yakovlev, V.A. Effect of the Ni/Cu Ratio on the Composition and Catalytic Properties of Nickel-Copper Alloy in Anisole Hydrodeoxygenation. Kinet Catal 2014, 55, 69–78, doi:10.1134/S0023158414010145.
Reviewer 2 Report
This paper is relevant to the catalytic conversion of renewable bio-derived resources to value-added chemicals. More specifically, it concerns the hydrogenation of furfural to furfuryl alcohol (FA), a valuable commodity chemical, over a new Cu-Fe-Al/SiO2 catalyst prepared by the sol-gel method. The new catalyst shows a very high efficiency, providing 97–99% FA yield at a 100% furfural conversion at 100–140oC. The catalyst is sufficiently characterised by a range of techniques. Overall, this work has achieved considerable progress in furfural hydroconversion and will be of interest to the readership of this journal. I recommend acceptance of this paper for publication after rather minor corrections as detailed below.
1) Adsorption of CO reflects the number of accessible Cu sites, not ‘the number of surface particles’ (p. 2).
2) What proves that the addition of Fe reduces Cu particle size? This should be made clear in the text.
3) What is the meaning of the numbers shown in HRTEM images (Fig. 9 and Fig. 10)? This should be explained.
4) When optimising the catalyst performance, the authors often compare very close values of conversion, yield, etc. To make this meaningful the experimental error should be reported.
5) In the figure and table legends, catalyst amounts should be given as a percentage rather than weight.
6) The abbreviation CSR (p. 4) needs explaining.
7) In Table 5, whole numbers and fractions should be separated by the decimal point (dot), not by the comma.
8) Finally, extensive editing of the English language and style is required.
Author Response
Reviewer #2
This paper is relevant to the catalytic conversion of renewable bio-derived resources to value-added chemicals. More specifically, it concerns the hydrogenation of furfural to furfuryl alcohol (FA), a valuable commodity chemical, over a new Cu-Fe-Al/SiO2 catalyst prepared by the sol-gel method. The new catalyst shows a very high efficiency, providing 97–99% FA yield at a 100% furfural conversion at 100–140oC. The catalyst is sufficiently characterised by a range of techniques. Overall, this work has achieved considerable progress in furfural hydroconversion and will be of interest to the readership of this journal. I recommend acceptance of this paper for publication after rather minor corrections as detailed below.
Response
Thank you for your interest of our work and your valuable comments helping us to improve this paper. A detailed response to each comment is provided below.
- Adsorption of CO reflects the number of accessible Cu sites, not ‘the number of surface particles’ (p. 2).
Response
According to your recommendation, “the number of surface particles” is changed to “the number of accessible Cu sites”. Please find the revised version of manuscript.
- What proves that the addition of Fe reduces Cu particle size? This should be made clear in the text.
Response
We have added an explanation and more discussed this effect in the text. The addition of iron increases the surface area of the catalysts (HetSG series) and prevents the crystallization of copper oxide, which is confirmed by XRD data (see Table 2). But it was not possible to determine the CSR of copper oxide due to its X-ray amorphism. Please find the revised version of manuscript.
- What is the meaning of the numbers shown in HRTEM images (Fig. 9 and Fig. 10)? This should be explained.
Response
The numbers indicate the interplanar distances of the indicated phases (nm). We have corrected these numbers and added an explanation in the captures to Figure 9 and 10. Please find the revised version of manuscript.
- When optimising the catalyst performance, the authors often compare very close values of conversion, yield, etc. To make this meaningful the experimental error should be reported.
Response
According to your recommendation, we have added the data on experimental error in the Section 3.4 (Product Analysis). Please find the revised version of manuscript.
- In the figure and table legends, catalyst amounts should be given as a percentage rather than weight.
Response
According to your recommendation, we have supplemented the description of the experimental technique (Section 3.3). The phrase (that corresponds to furfural/catalyst mass ratio of 16.2 or 4.9%, depending on the initial mass of catalyst.) has been added. Please find the revised version of manuscript.
- The abbreviation CSR (p. 4) needs explaining.
Response
We have added a description for CSR abbreviation in Section 2.2. The phrase (coherent scattering region) has been added. Please find the revised version of manuscript.
- In Table 5, whole numbers and fractions should be separated by the decimal point (dot), not by the comma.
Response
According to your recommendation, we have checked all numbers and corrected them. Please find the revised version of manuscript.
- Finally, extensive editing of the English language and style is required.
Response
According to your recommendation, English language in the manuscript was thoroughly checked and edited. The revised version of manuscript was checked using AJE Grammar Check tool (https://www.aje.com/grammar-check/). This tool shows 7.8/10 score (90th percentile of papers submitted to AJE) and indicates that the manuscript is well written and does not need language editing.
Reviewer 3 Report
Comments on IJMS-2311846
The study is devoted to the investigation of copper-based catalysts for the hydrogenation of furfural. A series of multicomponent catalysts has been obtained via sol-gel synthesis, characterized by several techniques and further tested in the reaction of furfural hydrogenation under various conditions. The topic is no doubt of interest. However, the manuscript in the present form suffers from numerous disadvantages requiring a deep revision. First of all, the quality of the language is unfortunately inappropriate; grammar mistakes are visible even for me (not being a native speaker). Further comments on the manuscript are listed below.
1. Introduction, the 1st paragraph. The applications of FA should be emphasized here, as further comes the reasoning about the importance of the high selectivity in hydrogenation. What for do we actually need this compound? The same for 2-MF.
2. Introduction, the 2nd paragraph. The other important disadvantage of the copper chromite catalysts is the high toxicity of chromium salts. This pushes the industry to develop Cr-free Cu-based catalysts. It is important to mention here while listing the disadvantages of the existing solution.
3. Introduction, the 2nd paragraph. Chromium oxide is normally very stable, especially in the absence of strong acids/bases. Please provide a reference to a source, where the phenomenon is described.
4. Introduction, the 2nd paragraph, the coating of copper by Cr-containing particles - please provide references to support the statement.
5. Page 2, the 2nd paragraph. Strong adsorption of the reaction products on the external surface of the catalyst is generally understood as the effect of inhibition (poisoning by product). The inhibition usually leads to the decrease of the reaction rate, while the changes in selectivity are not caused (at least 'by default'). I suggest therefore to reformulate the sentence - or to explain, how does the adsorption cause the selectivity changes.
6. Page 2, the 3rd paragraph. ‘Esterification of alcohols and aldehydes’ sounds confusing, as an alcohol cannot be esterified by an aldehyde.
7. Page 2, the 3rd paragraph, “…90% at 97%...” - It is in fact not appropriate to judge on the activity by the yield solely, as the latter depends on the conditions (T, p(H2), time, feed-to-cat ratio etc...)
8. Page 2, the 3rd paragraph, “Alumina is known…” - This sentence is not connected with the previous one: how and why do you suddenly come from the iron oxide to the alumina?
9. Page 2, “Thus, the use of copper-containing…” - The intention described is to combine Cu with FexOy and Al2O3 to obtain the doping effect. Two facts are discussed: Fe3O4 is a redox-active catalyst, and the alumina is an acid catalyst. And yet it does not mean that the Cu-based catalysts can benefit from such a combination: copper is already very active in redox reactions, including the aldehydes reduction. I encourage authors to improve the substantiation.
10. Figure 1 and the other figs/tables – please clearly specify the feed-to-catalyst ratio throughout the text, as the simple indication of the weights is not appropriate – from the subscript one cannot figure out, what the ratio amounts to
11. Please provide the references to the articles, where the application of CO sorption for the evaluation of Cu dispersion is described
12. Table 1 and Figure 1 partially duplicate each other. I think the Figure is not necessary here and could be moved into the 'Supplementary materials' section
13. Table 1. It is not correct to bring in comparison such the high yields. Normally taken for the purpose of comparison are the values of conversion/yields at short reaction times, where the reaction rate is closer to the 'initial reaction rate'. It is generally agreed that they should not exceed 20-25%. Please restructure the table with regard on this issue.
14. From the Fig. 3 one can see that it is the 40Cu-SIO2 that is the most active catalyst. Only the 37Cu7Fe one shown comparable reaction rate. At the same time, other catalysts afford somewhat higher yields at greater reaction times, but it could not be correctly explained by the "higher activity".
15. From Fig. 4 I cannot see that the doping of Cu-Fe catalysts with alumina allow the activity increase. Please check.
16. Please disclose in the text, what does the abbreviation 'CSR' stand for
17. Section 3.2, the 1st paragraph - Pasty is a small pie stuffed with a filling, like 'piroshki' are :-)
18. Page 14, the 1st line - X and Y are the theoretical values or the experimental ones obtained by the elemental analysis? Please specify
19. Section 3.3 - Were the catalytic tests duplicated/triplicated, as it is generally perforemd? Please disclose the data on the reproducibility
20. Section 3.3 – the word ‘plant’ is not appropriate
21. Section 3.3 - What time has it taken to reach the set temperatures? Please specify both for 100 and 250oC
22. Section 3.4 – better use ‘GC/MS’ instead of ‘chromato-mass spectrometer’, as the latter name is used solely in the Russian-speaking literature and is not used in the English language
23. Section 3.4, the GC/MS analysis - Carrier gas? Injector conditions? Ion source conditions? Column length, ID, film thickness?
24. Section 3.4, the GC analysis – which of the two columns was used here?
25. Section 3.4 - Please specify the procedure for the quantification. Normalization/IS method was used? Which standard was used? How the response factors were determined?
26. Section 3.4 - Retention time is not sufficient for the unambiguous identification by itself; it needs additional methods. The retention times and retention indices should be supported by something. Moreover, authors used a low-polar column for the qualitative analysis and a polar column for the quantitative one. The orders of elution (and the RI's) differ between these two columns; perhaps the carrier gases are also different. Please describe how the qualitative analysis was performed.
I encourage the authors to undertake the recommended revision and to have a closer look at the manuscript in general, as in fact the comments listed reflect only the general issues. In fact, the manuscript could be accepted only after a deep revision. I also recommend the 2nd review in order to check whether the authors could improve both the flow and the content.
Author Response
Reviewer #3
The study is devoted to the investigation of copper-based catalysts for the hydrogenation of furfural. A series of multicomponent catalysts has been obtained via sol-gel synthesis, characterized by several techniques and further tested in the reaction of furfural hydrogenation under various conditions. The topic is no doubt of interest. However, the manuscript in the present form suffers from numerous disadvantages requiring a deep revision. First of all, the quality of the language is unfortunately inappropriate; grammar mistakes are visible even for me (not being a native speaker). Further comments on the manuscript are listed below.
Response
Thank you for your interest of our work and your valuable comments, helping us to improve this paper. A detailed response to each comment is provided below.
According to your recommendation, English language in the manuscript was thoroughly checked and edited. The revised version of manuscript was checked using AJE Grammar Check tool (https://www.aje.com/grammar-check/). This tool shows 7.8/10 score (90th percentile of papers submitted to AJE) and indicates that the manuscript is well written and does not need language editing.
- Introduction, the 1st paragraph. The applications of FA should be emphasized here, as further comes the reasoning about the importance of the high selectivity in hydrogenation. What for do we actually need this compound? The same for 2-MF.
Response
According to your recommendation, we have added an information on the application of FA and 2-MF compounds. Several relevant references have been added. Please find the revised version of manuscript.
- Yan, K.; Chen, A. Efficient Hydrogenation of Biomass-Derived Furfural and Levulinic Acid on the Facilely Synthesized Noble-Metal-Free Cu–Cr Catalyst. Energy 2013, 58, 357–363, doi:10.1016/j.energy.2013.05.035.
- Sulmonetti, T.P.; Hu, B.; Ifkovits, Z.; Lee, S.; Agrawal, P.K.; Jones, C.W. Vapor Phase Hydrogenolysis of Furanics Utilizing Reduced Cobalt Mixed Metal Oxide Catalysts. ChemCatChem 2017, 9, 1815–1823, doi:10.1002/cctc.201700228.
- Date, N.S.; Hengne, A.M.; Huang, K.-W.; Chikate, R.C.; Rode, C.V. Single Pot Selective Hydrogenation of Furfural to 2-Methylfuran over Carbon Supported Iridium Catalysts. Green Chem. 2018, 20, 2027–2037, doi:10.1039/C8GC00284C.
- Varila, T.; Mäkelä, E.; Kupila, R.; Romar, H.; Hu, T.; Karinen, R.; Puurunen, R.L.; Lassi, U. Conversion of Furfural to 2-Methylfuran over CuNi Catalysts Supported on Biobased Carbon Foams. Catalysis Today 2021, 367, 16–27, doi:10.1016/j.cattod.2020.10.027.
- Corma, A.; de la Torre, O.; Renz, M.; Villandier, N. Production of High-Quality Diesel from Biomass Waste Products. Angewandte Chemie International Edition 2011, 50, 2375–2378, doi:10.1002/anie.201007508.
- Corma, A.; Torre, O. de la; Renz, M. Production of High Quality Diesel from Cellulose and Hemicellulose by the Sylvan Process: Catalysts and Process Variables. Energy Environ. Sci. 2012, 5, 6328–6344, doi:10.1039/C2EE02778J.
- Zeitsch, K.J. The Chemistry and Technology of Furfural and Its Many By-Products; Elsevier, 2000; ISBN 978-0-08-052899-1.
- Taylor, M.J.; Durndell, L.J.; Isaacs, M.A.; Parlett, C.M.A.; Wilson, K.; Lee, A.F.; Kyriakou, G. Highly Selective Hydrogenation of Furfural over Supported Pt Nanoparticles under Mild Conditions. Applied Catalysis B: Environmental 2016, 180, 580–585, doi:10.1016/j.apcatb.2015.07.006.
- Yan, K.; Wu, G.; Lafleur, T.; Jarvis, C. Production, Properties and Catalytic Hydrogenation of Furfural to Fuel Additives and Value-Added Chemicals. Renewable and Sustainable Energy Reviews 2014, 38, 663–676.
- Wang, Y.; Zhao, D.; Rodríguez-Padrón, D.; Len, C. Recent Advances in Catalytic Hydrogenation of Furfural. Catalysts 2019, 9, 796, doi:10.3390/catal9100796.
- Schneider, M.H.; Phillips, J.G. Furfuryl Alcohol and Lignin Adhesive Composition 2004.
- Barr, J.B.; Wallon, S.B. The Chemistry of Furfuryl Alcohol Resins. Journal of Applied Polymer Science 1971, 15, 1079–1090, doi:10.1002/app.1971.070150504.
- Recent Advances in the Catalytic Transfer Hydrogenation of Furfural to Furfuryl Alcohol over Heterogeneous Catalysts. Green Chemistry 2022, 24, 1780–1808, doi:10.1039/d1gc04440k.
- Vaidya, P.D.; Mahajani, V.V. Kinetics of Liquid-Phase Hydrogenation of Furfuraldehyde to Furfuryl Alcohol over a Pt/C Catalyst. Ind. Eng. Chem. Res. 2003, 42, 3881–3885, doi:10.1021/ie030055k.
- Sitthisa, S.; Sooknoi, T.; Ma, Y.; Balbuena, P.B.; Resasco, D.E. Kinetics and Mechanism of Hydrogenation of Furfural on Cu/SiO2 Catalysts. Journal of Catalysis 2011, 277.
- Introduction, the 2nd paragraph. The other important disadvantage of the copper chromite catalysts is the high toxicity of chromium salts. This pushes the industry to develop Cr-free Cu-based catalysts. It is important to mention here while listing the disadvantages of the existing solution.
Response
Some examples of existing solutions to overcome the disadvantages of chromium-containing catalysts are given in paragraph 2 (for example, the use of atomic layer deposition of alumina, alternative catalysts based on noble / transition metals). We have placed an emphasis on this point. Please find the revised version of manuscript.
- Introduction, the 2nd paragraph. Chromium oxide is normally very stable, especially in the absence of strong acids/bases. Please provide a reference to a source, where the phenomenon is described.
Response
Introduction describes the existing copper-chromium catalysts for the industrial hydrogenation of furfural. These catalysts make it possible to obtain furfuryl alcohol with a high yield, but they have some disadvantages. The ways are given to solve the problems associated with decontamination and provides alternative catalysts for the furfural hydrotreatment. Relevant references are provided. In the revised version, we have added more references. Please find the revised manuscript.
- Introduction, the 2nd paragraph, the coating of copper by Cr-containing particles - please provide references to support the statement.
Response
According to your recommendation, we have added several references concerning the deactivation Cu-Cr catalysts (e.g., coating of copper by Cr-containing particles). Please find the revised version of manuscript.
- Zhang, H.; Lei, Y.; Kropf, A.J.; Zhang, G.; Elam, J.W.; Miller, J.T.; Sollberger, F.; Ribeiro, F.; Akatay, M.C.; Stach, E.A.; et al. Enhancing the Stability of Copper Chromite Catalysts for the Selective Hydrogenation of Furfural Using ALD Overcoating. Journal of Catalysis 2014, 317, 284–292, doi:10.1016/j.jcat.2014.07.007.
- Liu, D.; Zemlyanov, D.; Wu, T.; Lobo-Lapidus, R.J.; Dumesic, J.A.; Miller, J.T.; Marshall, C.L. Deactivation Mechanistic Studies of Copper Chromite Catalyst for Selective Hydrogenation of 2-Furfuraldehyde. Journal of Catalysis 2013, 299, 336–345, doi:10.1016/j.jcat.2012.10.026.
- Page 2, the 2nd paragraph. Strong adsorption of the reaction products on the external surface of the catalyst is generally understood as the effect of inhibition (poisoning by product). The inhibition usually leads to the decrease of the reaction rate, while the changes in selectivity are not caused (at least 'by default'). I suggest therefore to reformulate the sentence - or to explain, how does the adsorption cause the selectivity changes.
Response
According to your recommendation, we have changed some statements about the adsorption of reaction products on the external surface of the catalyst. Please find the revised version of manuscript.
- Page 2, the 3rd paragraph. ‘Esterification of alcohols and aldehydes’ sounds confusing, as an alcohol cannot be esterified by an aldehyde.
Response
The formation of esters by oxidative esterification from aldehydes in the presence of alcohols (e.g, methanol, ethanol) is described in the paper [Rajabi, F.; Arancon, R.A.D.; Luque, R. Oxidative Esterification of Alcohols and Aldehydes Using Supported Iron Oxide Nanoparticle Catalysts. Catalysis Communications 2015, 59, 101–103, doi:10.1016/j.catcom.2014.09.022.] cited in the manuscript.
- Page 2, the 3rd paragraph, “…90% at 97%...” - It is in fact not appropriate to judge on the activity by the yield solely, as the latter depends on the conditions (T, p(H2), time, feed-to-cat ratio etc...)
Response
The details on process conditions have been added to the text (e.g., 160 °C, 5 h, 0.1 g of catalyst, 0.67 mmol of furfural, 20ml of alcohol). Please find the revised version of manuscript.
- Page 2, the 3rd paragraph, “Alumina is known…” - This sentence is not connected with the previous one: how and why do you suddenly come from the iron oxide to the alumina?
Response
One of the ways to increase the activity of copper catalysts is application of some additives (e.g., Fe, Mo, Al, Ni, Co, Zn). This paragraph is focused on the modifiers, which are used in this study (i.e., Fe and Al). We have added a discussion between the parts describing these modifiers. Please find the revised version of manuscript.
- Page 2, “Thus, the use of copper-containing…” - The intention described is to combine Cu with FexOy and Al2O3 to obtain the doping effect. Two facts are discussed: Fe3O4 is a redox-active catalyst, and the alumina is an acid catalyst. And yet it does not mean that the Cu-based catalysts can benefit from such a combination: copper is already very active in redox reactions, including the aldehydes reduction. I encourage authors to improve the substantiation.
Response
The monometallic copper catalysts are known to be highly active in the hydrogenation of furfural, but there are a few disadvantages in their application (they are mentioned in Introduction of the manuscript). The main issue is the formation of carbon deposits. Therefore, the promising approach in this field is the modification of copper catalysts with various metals, which can lead to an improvement in its catalytic properties (the relevant references are cited in the text). In this work, for the first time, we synthesized catalysts by the heterophase sol-gel method and showed their improved activity in the process of furfural hydrogenation compared to the monocopper sample. The choice of modifiers is justified in the introduction with relevant references.
- Figure 1 and the other figs/tables – please clearly specify the feed-to-catalyst ratio throughout the text, as the simple indication of the weights is not appropriate – from the subscript one cannot figure out, what the ratio amounts to
Response
According to your recommendation, we have supplemented the description of the experimental technique (Section 3.3). The phrase (that corresponds to furfural/catalyst mass ratio of 16.2 or 4.9%, depending on the initial mass of catalyst) has been added. Please find the revised version of manuscript.
- Please provide the references to the articles, where the application of CO sorption for the evaluation of Cu dispersion is described
Response
According to your recommendation, we have added some references about CO chemisorption on Cu catalysts. Please find the revised version of manuscript.
- Parris, G.E.; Klier, K. The Specific Copper Surface Areas in CuZnO Methanol Synthesis Catalysts by Oxygen and Carbon Monoxide Chemisorption: Evidence for Irreversible CO Chemisorption Induced by the Interaction of the Catalyst Components. Journal of Catalysis 1986, 97, 374–384, doi:10.1016/0021-9517(86)90009-6.
- Phillips, J.M.; Leibsle, F.M.; Holder, A.J.; Keith, T. A Comparative Study of Chemisorption by Density Functional Theory, Ab Initio, and Semiempirical Methods: Carbon Monoxide, Formate, and Acetate on Cu(110). Surface Science 2003, 545, 1–7, doi:10.1016/j.susc.2003.07.006.
- Smirnov, A.A.; Khromova, S.A.; Bulavchenko, O.A.; Kaichev, V.V.; Saraev, A.A.; Reshetnikov, S.I.; Bykova, M.V.; Trusov, L.I.; Yakovlev, V.A. Effect of the Ni/Cu Ratio on the Composition and Catalytic Properties of Nickel-Copper Alloy in Anisole Hydrodeoxygenation. Kinet Catal 2014, 55, 69–78, doi:10.1134/S0023158414010145.
- Table 1 and Figure 1 partially duplicate each other. I think the Figure is not necessary here and could be moved into the 'Supplementary materials' section.
Response
We believe that both Figure 1 and Table 1 are important to the main section of manuscript due to the different semantic load. Thus, the figure clearly demonstrates the behavior of catalysts with different copper content on the reaction time. The table shows the correlations between the amount of chemisorbed carbon monoxide and the yield of FA.
- Table 1. It is not correct to bring in comparison such the high yields. Normally taken for the purpose of comparison are the values of conversion/yields at short reaction times, where the reaction rate is closer to the 'initial reaction rate'. It is generally agreed that they should not exceed 20-25%. Please restructure the table with regard on this issue.
Response
We agree that it’s not correct to bring in comparison the high yields, but our catalysts have similar FA yields at short reaction times (except 5Cu and 50Cu). Furthermore, one of the tasks of this work was to obtain a catalyst, which is able to provide high values of FA yield. Thus, we consider that it is reasonable to compare these values.
- From the Fig. 3 one can see that it is the 40Cu-SiO2 that is the most active catalyst. Only the 37Cu7Fe one shown comparable reaction rate. At the same time, other catalysts afford somewhat higher yields at greater reaction times, but it could not be correctly explained by the "higher activity".
Response
According to your comment, we have changed our reasoning about the activity of catalyst. We associate the increase in the FA yield with a change in the particle size of the active component, which is confirmed by XRD. Please find the revised version of manuscript.
- From Fig. 4 I cannot see that the doping of Cu-Fe catalysts with alumina allow the activity increase. Please check.
Response
The CuFe-SiO2 catalyst is active in the hydrogenation of furfural. Furfural conversion over this catalyst achieved 99%. In the case of the CuFeAl-SiO2 catalyst, the furfural conversion was 100%, and only alcohol was detected as product. Therefore, we state that this catalyst is more active than CuFe-SiO2.
- Please disclose in the text, what does the abbreviation 'CSR' stand for
Response
The CSR abbreviation has been explained in the Section 2.2. The phrase (coherent scattering region) has been added. Please find the revised version of manuscript.
- Section 3.2, the 1st paragraph - Pasty is a small pie stuffed with a filling, like 'piroshki' are :-)
Response
We have changed “pasty” to “paste-like” in Section 3.2. Please find the revised version of manuscript.
- Page 14, the 1st line - X and Y are the theoretical values or the experimental ones obtained by the elemental analysis? Please specify.
Response
X and Y are the experimental values obtained by the elemental analysis. More detailed information on the ICP–AES methodology is presented in Section 3.5.
- Section 3.3 - Were the catalytic tests duplicated/triplicated, as it is generally perforemd? Please disclose the data on the reproducibility
Response
The catalytic tests were carried out repeatedly, and the data on furfural conversion and product selectivity were reproduced, considering the experimental error of product analysis (ca. 5%).
- Section 3.3 – the word ‘plant’ is not appropriate
Response
We have removed the word “plant” and changed the description of experimental setup in Section 3.3. Please find the revised version of manuscript.
- Section 3.3 - What time has it taken to reach the set temperatures? Please specify both for 100 and 250oC
Response
According to your recommendation, we have added an information on the heat rate (Next, the reactor was sealed, and the mixture was heated to the required temperature (100–250 °C) with a heat rate of 10 °C/min ) in Section 3.3. Please find the revised version of manuscript.
- Section 3.4 – better use ‘GC/MS’ instead of ‘chromato-mass spectrometer’, as the latter name is used solely in the Russian-speaking literature and is not used in the English language
Response
According to your recommendation, we have changed the name of this equipment.
- Section 3.4, the GC/MS analysis - Carrier gas? Injector conditions? Ion source conditions? Column length, ID, film thickness?
Response
According to your recommendation, we have added information about GC/MS analysis. Qualitative analysis of liquid reaction products was carried out on an Agilent 7000B GC/MS (Agilent Technologies Inc., Santa Clara, CA, USA) with a triple quadrupole analyzer and an HP-5 MS quartz capillary column ((5%-phenyl)-methylpolysiloxane, length of 30 m, inner diameter of 0.25 mm, phase thickness of 0.25 μm) from Agilent Technologies Inc (Santa Clara, CA, USA). The temperature program was as follows: 50 °C/min for 3 min, then 10 °C/min to 260 °C/min. Mass spectra recording conditions: electron ionization (70 eV), scanning mode in the m/z range of 40-500. Helium was used as a carrier gas. NIST.11. database was used to identify the components of the analyzed sample. Please find the revised version of manuscript.
- Section 3.4, the GC analysis – which of the two columns was used here?
Response
The СМ-Wax column was mainly used for the analysis of products. At high temperatures (180-250 °C) 2-methyltetrahydrofuran was formed, the peak of which overlaps with the peak of 2-methylfuran on the СМ-Wax column. In this case, the analysis was carried out on the HP-5 column.
- Section 3.4 - Please specify the procedure for the quantification. Normalization/IS method was used? Which standard was used? How the response factors were determined?
Response
We have specified the procedure for the quantification in Section 3.4. The product quantification was determined using the normalization method. Since the initial substance (furfural) and reaction products (furfuryl alcohol, 2-methylfuran and 2-methyltetrahydrofuran) are similar in their structure, the response factors are taken equal to unity. Please find the revised version of manuscript.
- Section 3.4 - Retention time is not sufficient for the unambiguous identification by itself; it needs additional methods. The retention times and retention indices should be supported by something. Moreover, authors used a low-polar column for the qualitative analysis and a polar column for the quantitative one. The orders of elution (and the RI's) differ between these two columns; perhaps the carrier gases are also different. Please describe how the qualitative analysis was performed.
Response
Additionally to quantitative analysis, qualitative analysis of liquid reaction products was carried out on an Agilent 7000B GC/MS, NIST.11. database was used to identify the components of the analyzed sample. More detailed information about the GC/MS methodology is presented in Section 3.4.
Round 2
Reviewer 3 Report
Dear Authors,
thank you very much for your elaborated answers to my comments. I agree that in the present form the manuscript became improved.
However there are still some points under consideration:
1. When answering the comment No. 15 you claim that the Al-doped catalyst is more active than the non-doped one since... the former gives 100% versus the 99% of the latter. This point of view is in fact fallacious. The activities should be compared by the initial reaction rates, by the half life time, by the reaction rate constants, by TOFs/TONs. Moreover, you claimed (when answering my comment No. 19) that your experimental error is about 5%. How do you come to the conclusion starting from two single results: 99±5% and 100±5%? In fact it is not supported by your experimental data.
2. Your quantification method (GC) is wrong, sorry for this word, but it is so. The response factor of a flame ionization detector (in the case of oxygenates) is proportional to the weight content of carbon. C/O ratio for furfural is 5/2, twice lower for 2-MF. I am absolutely convinced that the concentration you measure through your described GC methode are not the true concentrations. Moreover, if you review the literature you find that even compounds with similar molecular formula (e.g. propanol-1 and propanol-2) have inequal response factors when analyzed by GC-FID. Please either correct your GC methode or clearly state in the manuscript that the concentrations that you operate are not the weight concentrations, but (normalized) peak areas. I am sorry but I cannot admit this paper while it contains such a visible mistake, except you can prove that your method gives true concentrations.
3. Please add the description of the yield calculation. The yield in chemical kinetics is normally expressed in mol %, the wt% is more common in the industrial practice (e.g. recovery of the gasoline from the oil). Regarding Fig. 5 - the composition is expressed in molar or weight percent?
4. Please prepare two versions of the manuscript (1- changes marked, 2 - clear) for the next review round.
Author Response
Responses to reviewer’s comments
We would like to thank the reviewer for his in-depth analysis and constructive remarks, which helped us improve the manuscript. Please find the detailed answers below.
Comment 1. When answering the comment No. 15 you claim that the Al-doped catalyst is more active than the non-doped one since... the former gives 100% versus the 99% of the latter. This point of view is in fact fallacious. The activities should be compared by the initial reaction rates, by the half life time, by the reaction rate constants, by TOFs/TONs. Moreover, you claimed (when answering my comment No. 19) that your experimental error is about 5%. How do you come to the conclusion starting from two single results: 99±5% and 100±5%? In fact it is not supported by your experimental data.
Response:
We completely agree with the reviewer’s comment on incorrect use of “activity” term for comparing the mentioned catalysts, which led to a misunderstanding. It was due to a short form of answer. In our response to comment №15, we implied that the CuFeAl-SiO2 catalyst was selected for further investigation of the temperature effect based on considering the set of characteristics.
The complete conversion of furfural into the desired product (i.e., furfuryl alcohol) is the most important behavior. The amount of carbon deposits on the surface of catalyst after the reaction is also important parameter.
The CuFe-SiO2 and CuFeAl-SiO2 catalysts have similar catalytic characteristics (furfural conversion, FA yield), but the CuFeAl-SiO2 sample can have some advantages compared to the CuFe-SiO2 catalyst due to the physicochemical features related to the amount of carbon deposits and the phase composition after the reaction. We also showed using X-ray powder diffraction that the presence of iron reduces the dispersion of copper particles, while the introduction of a small amount of aluminum promotes their crystallization. According to HRTEM data, we suppose that iron and aluminum oxides are strongly covered with carbon deposits, while Cu-containing particles remain relatively free, which can make that catalyst effective in furfural hydrogenation. Based on all facts mentioned above, we selected CuFeAl-SiO2 catalyst for further investigation of the temperature effect on the distribution of products during the hydroconversion of furfural.
It is important to note that incorrect using of “activity” term was only in answer to reviewer’s comment, but the main text of manuscript contained correct comparison of CuFe-SiO2 and CuFeAl-SiO2 catalysts and detailed discussion on the reasons for selecting the latter one.
Finally, we can also illustrate the correct estimation of error in the yield (Y) of furfuryl alcohol (FA):
Please, find the formula in the attached PDF-file below
where is the initial amount of furfural (e.g., Please, find the formula in the attached PDF-file below), is the amount of formed FA (e.g., Please, find the formula in the attached PDF-file below)
Please, find the formula in the attached PDF-file below
where is the error of Y, is the error of (if we consider 5% of accuracy of GC method), is the error of ( , if we consider 5% of accuracy of GC method)
Please, find the formula in the attached PDF-file below
As a result
Please, find the formula in the attached PDF-file below
Or
Please, find the formula in the attached PDF-file below
Yes, these values are crossed, but, as explained above, we consider a set of characteristics for selecting the promising catalyst.
Comment 2. Your quantification method (GC) is wrong, sorry for this word, but it is so. The response factor of a flame ionization detector (in the case of oxygenates) is proportional to the weight content of carbon. C/O ratio for furfural is 5/2, twice lower for 2-MF. I am absolutely convinced that the concentration you measure through your described GC methode are not the true concentrations. Moreover, if you review the literature you find that even compounds with similar molecular formula (e.g. propanol-1 and propanol-2) have inequal response factors when analyzed by GC-FID. Please either correct your GC methode or clearly state in the manuscript that the concentrations that you operate are not the weight concentrations, but (normalized) peak areas. I am sorry but I cannot admit this paper while it contains such a visible mistake, except you can prove that your method gives true concentrations.
Response: There is no crude error in the performed calculations. The results of quantitative analysis using GC are valid because they are based on the first approximation of the effect of atomic composition on the response factor of substance. According to this approximation, the number of С atoms in substance for analysis is one of the main parameters, which strongly affect its response factor [M. Kallai and J. Balla. Chromatographia 56, 357–360 (2002); A.J. Andreatch and R.J. Cvetanovic, J. Gas Chromatogr., 32, 1021 (1960); V.G. Berezkin and V.S. Tatarinsky, Gas Chromatographic Methods of Impurity Analysis, "Nauka", Moscow 1970, p. 58].
The authors completely agree with the reviewer that usage of the exact values of the response factors is the most correct way to make quantitative analysis. Unfortunately, we cannot find in the literature these characteristics for the analyzed substances.
Here, we can provide a similar example described in the literature. It is the transformation of benzaldehyde [Katritzky, A.R.; Ignatchenko, E.S.; Barcock, R.A.; Lobanov, V.S.; Karelson, Mati. Prediction of Gas Chromatographic Retention Times and Response Factors Using a General Qualitative Structure-Property Relationships Treatment. Anal. Chem. 1994, 66, 1799–1807, Lučić, B.; Trinajstić, N.; Sild, S.; Karelson, M.; Katritzky, A.R. A New Efficient Approach for Variable Selection Based on Multiregression: Prediction of Gas Chromatographic Retention Times and Response Factors. J. Chem. Inf. Comput. Sci. 1999, 39, 610–621]:
Please, find the scheme in the attached PDF-file below
(*the response factors presented in parentheses are taken from the mentioned papers)
In this example, benzaldehyde and benzyl alcohol have the same response factors (i.e., 0.81 and 0.80, respectively). In our case, it correlates with the conversion of furfural to furfuryl alcohol that corresponds to ~90% of all presented results. Therefore, using the equal response factors for furfural and furfuryl alcohol is valid.
Returning to the example on benzaldehyde, a significant change in the response factor occurs in the transformation of benzyl alcohol to toluene with the removal of an oxygen atom (the corresponding response factor increases up to 1.08). In our case, this transformation correlates with the transformation of furfuryl alcohol to 2-methylfuran (2-MF). As mentioned above, we cannot find the information on difference in the response factors of furfuryl alcohol and 2-methylfuran (commonly, the researchers do not indicate the used values of the response factors in their publications [Wang, Z.; Wang, X.; Zhang, C.; Arai, M.; Zhou, L.; Zhao, F. Selective Hydrogenation of Furfural to Furfuryl Alcohol over Pd/TiH2 Catalyst. Molecular Catalysis 2021, 508, 111599, Date, N.S.; Hengne, A.M.; Huang, K.-W.; Chikate, R.C.; Rode, C.V. Single Pot Selective Hydrogenation of Furfural to 2-Methylfuran over Carbon Supported Iridium Catalysts. Green Chem. 2018, 20, 2027–2037, Fu, X.; Liu, Y.; Liu, Q.; Liu, Z.; Peng, Z. Preparation of Highly Active Cu/SiO2 Catalysts for Furfural to 2-Methylfuran by Ammonia Evaporation Method. Catalysts 2022, 12, 276])
To address this concern, we have prepared an equimolar mixture of furfural, furfuryl alcohol, 2-methylfuran, and 2-methyltetrahydrofuran to estimate the relative response factors (Rf). The obtained values are presented below, and they match with the trend for benzaldehyde described above.
Please, find the scheme in the attached PDF-file below
(*the estimated response factors are presented in parentheses)
Similarly to benzaldehyde and benzyl alcohol, Rf for furfural and furfuryl alcohol are very close (i.e., 1.03 and 1.00, respectively). As mentioned above, this transformation corresponds to ~90% of all experimental data. However, Rfs of 2-methylfuran and 2-methyltetrahydrofuran are found to be higher (i.e., 1.25 and 1.44, respectively). Based on these estimations, we have recalculated our data for the experiments, in which 2-methylfuran and 2-methyltetrahydrofuran are formed. Comparison between old and new results are shown in Table A below, and you can see that there is no substantial effect on the results for all temperatures.
Table A
|
|
180°C |
200°C |
250°C |
|||
|
|
old |
new (with Rf) |
old |
new (with Rf) |
old |
new (with Rf) |
|
Furfuryl alcohol |
95 |
96 |
93 |
94 |
9 |
12 |
|
2-methylfuran |
5 |
4 |
6 |
5 |
77 |
76 |
|
2-methyltetrahydrofuran |
0 |
0 |
2 |
1 |
14 |
12 |
Therefore, to be more correct in the revised version of manuscript, we presented new data recalculated using the estimated Rf values. According to these data, we have changed Figure 5. We have added specification for furfuryl alcohol, 2-methylfuran and 2-methyltetrahydrofuran in Section 3.1 and described the procedure for quantitative analysis in Section 3.4. Please find the revised version of manuscript:
Furfuryl alcohol (≥ 98%, Component Reactiv JSC, Moscow, Russia), 2-methylfuran (99%, Acros Organics, New Jersey, USA), and 2-methyltetrahydrofuran (≥ 99%, Acros Organics, New Jersey, USA) were used to determine the relative response factors.
The product quantification was determined using the normalization method with the relative response factors of 1.03, 1.00, 1.25, and 1.44 for furfural, furfuryl alcohol, 2-methylfuran, and 2-methyltetrahydrofuran, respectively.
Comment 3. Please add the description of the yield calculation. The yield in chemical kinetics is normally expressed in mol %, the wt% is more common in the industrial practice (e.g. recovery of the gasoline from the oil). Regarding Fig. 5 - the composition is expressed in molar or weight percent?
Response: We thank the reviewer for noticing this neglect in our paper. The relative yield of the reaction products was estimated as the molar ratio of the amount of formed product to the initial amount of furfural multiplied by 100%. Since we do not normalize the product content by molecular weight, the yield of products is expressed in mol.%. We have specified the procedure of quantification in Section 3.4. Please find the revised version of manuscript:
The relative yield of the reaction products (%) was estimated as the molar ratio of the amount of formed product to the initial amount of furfural multiplied by 100%.
Comment 4. Please prepare two versions of the manuscript (1- changes marked, 2 - clear) for the next review round.
Response: According to your recommendation, we have prepared two versions of the manuscript and uploaded the marked PDF version to the response form in the system.

Round 3
Reviewer 3 Report
The authors have improved the manuscript according to the reviewers' comments. In the present form in could be accepted for publication. I thank the authors for the undertaken job that allowed significant improvements of the manuscript.